## OPEN

# The NALCN channel regulates metastasis and nonmalignant cell dissemination

Eric P. Rahrmann[1], David Shorthouse [2], Amir Jassim[1], Linda P. Hu[1], Mariaestela Ortiz[3], Betania Mahler-Araujo[4], Peter Vogel [5], Marta Paez-Ribes[1], Atefeh Fatemi[1], Gregory J. Hannon [1], Radhika Iyer[6], Jay A. Blundon[7], Filipe C. Lourenço [1], Jonathan Kay [8], Rosalynn M. Nazarian[9], Benjamin A. Hall [2], Stanislav S. Zakharenko [7], Douglas J. Winton [1], Liqin Zhu [10] and Richard J. Gilbertson [1,11] ✉

We identify the sodium leak channel non-selective protein (NALCN) as a key regulator of cancer metastasis and nonmalignant cell dissemination. Among 10,022 human cancers, *NALCN* loss-of-function mutations were enriched in gastric and colorectal cancers. Deletion of *Nalcn* from gastric, intestinal or pancreatic adenocarcinomas in mice did not alter tumor incidence, but markedly increased the number of circulating tumor cells (CTCs) and metastases. Treatment of these mice with gadolinium—a NALCN channel blocker—similarly increased CTCs and metastases. Deletion of *Nalcn* from mice that lacked oncogenic mutations and never developed cancer caused shedding of epithelial cells into the blood at levels equivalent to those seen in tumor-bearing animals. These cells trafficked to distant organs to form normal structures including lung epithelium, and kidney glomeruli and tubules. Thus, NALCN regulates cell shedding from solid tissues independent of cancer, divorcing this process from tumorigenesis and unmasking a potential new target for antimetastatic therapies.

Most patients with cancer die as a result of metastasis[1], the process by which cancer cells spread from the primary tumor to other organs in the body[2]. Blocking metastasis could markedly improve the survival of patients with cancer, but how this process is triggered within the complex cascade of tumorigenesis remains unclear[3].

Because metastasis is thought to be a wholly abnormal process, restricted to malignant tissues, attention has focused on identifying genetic mutations as drivers of cancer metastasis. Although this research has unmasked genes that promote metastasis in mouse models and humans, including a variety of ion channels that induce a metastasis-like phenotype by altering the transmembrane voltage to induce changes in gene transcription[4–6], so far no recurrent metastasis-specific mutations have been identified[2,3,7].

Other cell functions implicated in the metastatic cascade include 'stem cell-like' multipotency and plasticity. Stem cell capacity has been ascribed to metastatic cancer cells because of their ability to reconstitute heterogenous malignant cell populations as metastatic tumors[8,9]. Epithelial mesenchymal transition (EMT)[2]—a type of cellular plasticity displayed during normal gastrulation and tissue healing—is also an established feature of the metastatic cascade[2,10]. What remains unclear is how cancers 'hijack' these normal cell functions to enable metastasis.

Here, we identify a single ion channel, NALCN, as a key regulator of epithelial cell trafficking to distant tissues. NALCN is responsible for the background sodium leak conductance that maintains the resting membrane potential. It regulates key functions in excitable tissues, for example, respiration and circadian rhythms[11–13], and gain-of-function mutations in the gene are associated with neurological disorders[14]. However, little is known about the role of NALCN in nonexcitable tissues. We show that NALCN regulates the release of malignant and normal epithelial cells into the blood, and their trafficking to distant sites where they form metastatic cancers, or apparently normal tissues, respectively. We thereby demonstrate that the metastatic cascade can be triggered and operate independent of tumorigenesis. These observations have profound implications for understanding epithelial cell trafficking in health and disease and identify a novel target for antimetastatic therapies.

## Results

**NALCN loss-of-function in cancer.** We showed previously that Prominin1 (PROM1) marks basal stem cells in gastric antral glands and that their lineage forms adenocarcinomas in *Prom1^{CreERT2/LacZ}; Kras^{G12D};Trp53^{Flx/Flx}* (*P1^{KP}*) mice[15]. PROM1+, but not PROM1−, cells isolated from *P1^{KP}* gastric adenocarcinomas (*P1^{KP}*-GAC) propagated these tumors as allografts, suggesting that PROM1+ *P1^{KP}*-GAC cells are the malignant counterparts of antral gland basal stem cells (Extended Data Fig. 1). To understand how antral gland basal stem cells are corrupted during transformation, we compared their transcriptomes with those of PROM1+ *P1^{KP}*-GAC cells. Ion channels

[1]Cancer Research UK Cambridge Institute, University of Cambridge, Cambridge, UK. [2]Department of Medical Physics and Biomedical Engineering, University College London, London, UK. [3]Molecular Pharmacology Lab, Organoid Models Research and Biology, National Cancer Institute, Leidos Biomedical Research, Frederick, MD, USA. [4]Wellcome-MRC Institute of Metabolic Science, Histopathology Core, Cambridge University Hospitals NHS Foundation Trust, Cambridge, UK. [5]Veterinary Pathology Core Laboratory, St Jude Children's Research Hospital, Memphis, TN, USA. [6]Texas Children's Cancer and Hematology Centers, Houston, TX, USA. [7]Department of Developmental Neurobiology, St Jude Children's Research Hospital, Memphis, TN, USA. [8]Departments of Medicine and of Population and Quantitative Health Sciences, University of Massachusetts Medical School and UMass Memorial Medical Center, Worcester, MA, USA. [9]Massachusetts General Hospital, Pathology Service, Dermatopathology Unit, Boston, MA, USA. [10]Department of Pharmaceutical Sciences, St Jude Children's Research Hospital, Memphis, TN, USA. [11]Department of Oncology, University of Cambridge, Cambridge, UK. ✉e-mail: Richard.Gilbertson@cruk.cam.ac.uk

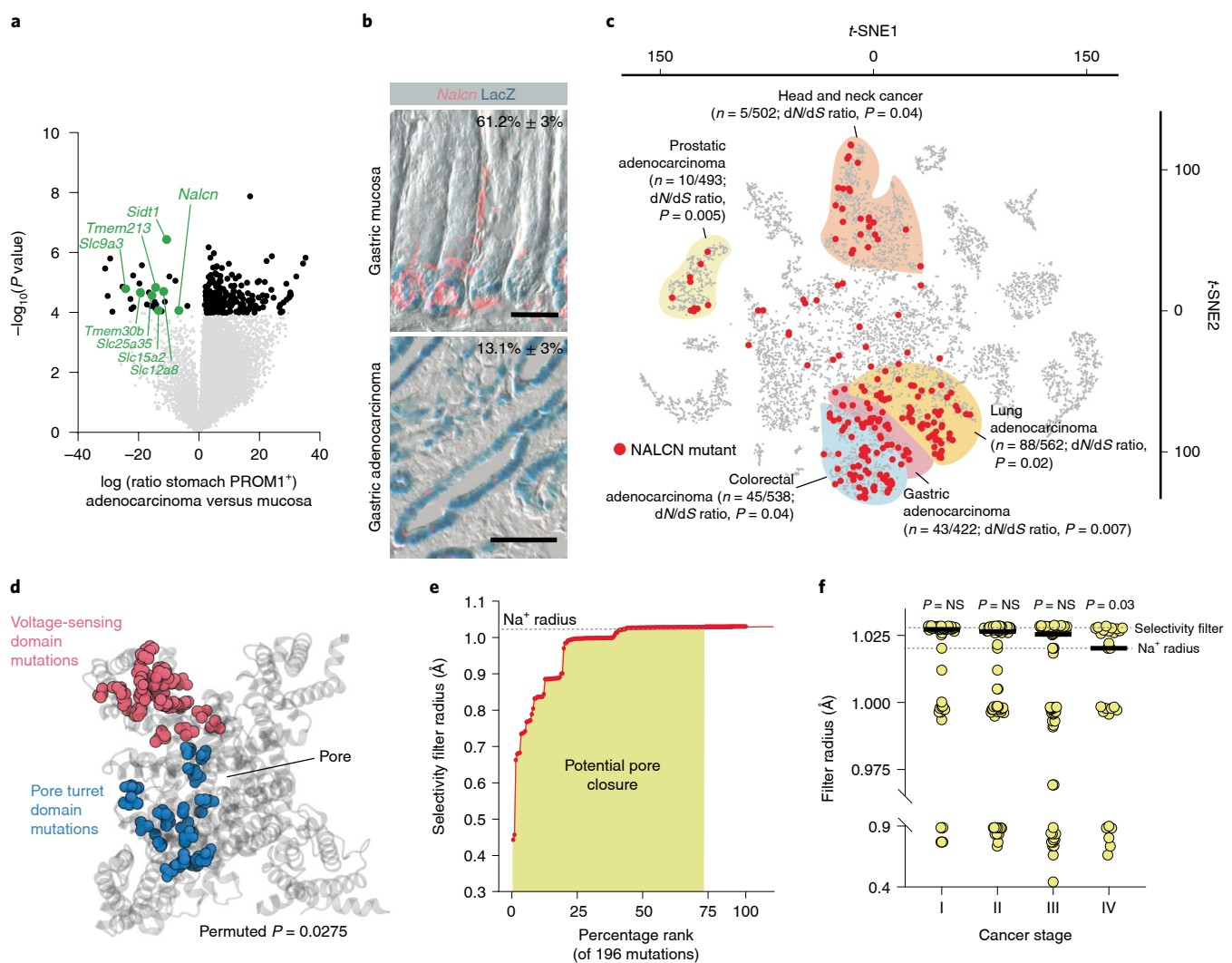

**Fig. 1 | NALCN loss-of-function in aggressive cancers. a**, Differential gene expression between normal PROM1+ gastric cells and *P1^KP*-GAC cells (downregulated ion channels are highlighted). Benjamini–Hochberg corrected *P* value, alpha=0.05. **b**, *Nalcn* RNA in situ hybridization and Prom1 expression (β-galactosidase (LacZ)) in *Prom1^CreERT2/LacZ* mouse stomach (*n*=3 biological replicates, 10 fields each; upper) and *P1^KP*-GAC (*n*=3 biological replicates, 10 fields each; lower). Numbers are shown as mean ± s.e.m. *Prom1+/Nalcn+* cells. Scale bar, 50 μm. **c**, *t*-SNE plot of 10,022 human cancers (*P* value, dN/dS shown; Source data). **d**, Mutant residues enriched in NALCN pore turret (blue) and voltage-sensing (red) domains. *P*=0.0275; permuted *P* value for probability of observing two clusters of 20 and 25 residues. **e**, Impact of 196 *NALCN* mutations on selectivity filter radius determined by HOLE analysis. **f**, NALCN pore closure by *NALCN* mutations in stage I (*n*=47), stage II (*n*=73), stage III (*n*=74) and stage IV (*n*=27) human cancers. Two-tailed Mann–Whitney *U*-test: stage I versus II, *P*=0.3488; stage I versus III, *P*=0.1613; stage I versus IV, *P*=0.0293. Bar denotes median filter radius.

and solute carriers were selectively downregulated in PROM1+ *P1^KP*-GAC cells (Fig. 1a and Supplementary Table 1). Among these, NALCN—a leak sodium channel that contributes to the resting cell membrane potential and cell excitability[13,16,17]—was restricted in its expression to PROM1+ antral gland basal stem cells and downregulated in PROM1+ *P1^KP*-GAC cells (Fig. 1b). Among 10,022 human cancers within The Cancer Genome Atlas, nonsynonymous mutations in *NALCN* were enriched in gastric, colorectal, lung, prostate and head and neck cancers (Fig. 1c)[18,19]. These cancers also contained deletions, and nonsense and frameshift mutations, at a frequency very similar to those observed in *TP53* in human cancer[20], suggesting *NALCN* might be a tumor suppressor (Supplementary Table 2).

To determine how nonsynonymous mutations might affect NALCN function in cancer, we used HOLE analysis[21] to predict their impact on the ion channel pore radius of NALCN embedded and relaxed within a 575-lipid 1-palmitoyl-2-oleoyl-*sn*-glycero-3

-phosphocholine bilayer in silico[12,22,23]. This model correctly predicted opening of the NALCN channel by 22 mutations known to confer gain-of-function[12], and closure of the channel by two mutations that cause loss-of-function[11] (Supplementary Table 3). Nonsynonymous, cancer-associated mutations were clustered within the pore turret and voltage-sensing domains that regulate NALCN channel opening[11,12]: 75% (*n*=147/196) of these mutations were predicted to close the channel (Fig. 1d,e and Supplementary Table 4). Mutations predicted to cause the greatest pore closure were enriched in the most advanced cancers (Fig. 1f). Furthermore, human GACs in which *NALCN* was mutated, upregulated genes associated with EMT[24], metastasis and cell migration (Supplementary Tables 5 and 6).

As a first step to test whether *Nalcn* regulates cancer progression, we altered its function in *P1^KP*-GAC cells using genetic (*Nalcn*-short hairpin RNA and *NALCN*-complementary DNA lentiviral transduction) or chemical (gadolinium chloride; GdCl₃)[13] approaches.

Whole-cell voltage-clamp analysis of $P1^{KP}$-GAC cells showed a linear $GdCl_3$-sensitive current to voltage steps in the $\pm 80$ mV range, as previously reported (Fig. 2a,b)[13]. Decreasing *Nalcn* expression in $P1^{KP}$-GAC cells eliminated the NALCN current, increased cell proliferation and conferred an EMT morphology and transcriptome on $P1^{KP}$-GAC orthotopic allografts (Fig. 2 and Supplementary Tables 7,8). Conversely, increased *Nalcn* expression increased the $GdCl_3$-sensitive current in $P1^{KP}$-GAC cells, decreased cell proliferation and conferred a hyperepithelialized morphology on allografts.

**Loss of Nalcn promotes cancer metastasis.** To study how *Nalcn* loss-of-function impacts cancer initiation and progression in intact tissues, we generated mice harboring a conditional *Nalcn* allele (*Nalcn^Flx*; Extended Data Fig. 2). These mice were bred with $P1^{KP}$, *Villin1-Cre^{ERT2};Kras^{G12D};Trp53^{Flx/Flx}* ($V1^{KP}$) or *Pdx1-Cre;Kras^{G12D};Trp53^{Flx/+}* ($Pdx1^{KP}$) mice to produce equivalent numbers of male and female mice that were either *Nalcn* wild-type (*Nalcn^{+/+}*), *Nalcn^{+/Flx}* or *Nalcn^{Flx/Flx}* (total $n = 551$; Supplementary Table 9). All mice carried the *Rosa26-ZsGreen* (*Rosa26^{ZSG}*) lineage-tracing allele. Cancers in $V1^{KP}$ and $Pdx1^{KP}$ mice are restricted by Cre expression to the intestine[25,26] and pancreas[27,28], respectively. *Prom1^{CreERT2/LacZ}* is expressed by a variety of stem/progenitor cells and induces tumors of the small intestine, liver, lung, salivary glands, prostate, uterus, skin and stomach in $P1^{KP}$ mice[15,29]. Because tissues can display age-dependent susceptibility to transformation[15] we activated Cre-recombination in $P1^{KP}$ and $V1^{KP}$ mice using tamoxifen at postnatal day 3 (P3) or P60. As expected, $V1^{KP}$ ($n = 127/141$) and $Pdx1^{KP}$ ($n = 55/55$) mice developed intestinal and pancreatic tumors, respectively, whereas $P1^{KP}$ mice developed tumors in the stomach ($n = 49/269$), small intestine ($n = 59/269$) and other sites ($n = 108/269$)[15,26,28]; 99% ($n = 212/214$) of tumors in $P1^{KP}$ mice occurred as single primary tumor (Fig. 3a–g and Supplementary Table 9). Detailed macro- and microscopic analysis of tumors revealed no significant impact of age of induction, sex and/or *Nalcn* status on tumor incidence, type, tumor-free survival, tumor growth rate, immune cell infiltration, proliferation or other key primary tumor characteristics (Fig. 3, Extended Data Fig. 3a–c and Supplementary Tables 9–11). However, the transcriptomes of $P1^{KP}$-GAC and $Pdx1^{KP}$ pancreatic adenocarcinomas ($Pdx1^{KP}$-PACs) were enriched for genes associated with human CTCs and EMT (Fig. 3j).

In keeping with these transcriptomic changes, deletion of *Nalcn* dramatically increased cancer metastasis in $P1^{KP}$, $V1^{KP}$ and $Pdx1^{KP}$ mice (Fig. 4a–d, Extended Data Fig. 4 and Supplementary Table 12). Metastatic and primary tumors were distinguished from one another by combined histology review, cosegregation of 'matched' primary and secondary tumor transcriptomes by unsupervised hierarchical clustering, and enrichment of histology-predicted primary tumor gene sets within metastatic tumor transcriptomes (Fig. 4a,c and Extended Data Fig. 4). $V1^{KP}$ intestinal adenocarcinomas ($V1^{KP}$-IACs, $n = 27$ mice) and $Pdx1^{KP}$-PACs ($n = 19$ mice) in *Nalcn^{+/+}*

mice, produced $2.82 \pm 4.88$ (mean $\pm$ s.e.m.) and $5.53 \pm 4.02$ metastases per mouse, respectively (Fig. 4d and Supplementary Tables 9 and 12). In stark contrast, these same tumors in $V1^{KP};Nalcn^{+/Flx}$ ($n = 51$), $V1^{KP};Nalcn^{Flx/Flx}$ ($n = 26$), $Pdx1^{KP};Nalcn^{+/Flx}$ ($n = 23$) and $Pdx1^{KP};Nalcn^{Flx/Flx}$ ($n = 13$) mice, produced $16.82 \pm 5.69$ (two-tailed Mann–Whitney $U$-test, $P = 0.03$ relative to *Nalcn^{+/+}*), $26.04 \pm 10.18$ ($P = 0.0009$), $15.04 \pm 3.62$ ($P = 0.007$) and $13.46 \pm 5.01$ ($P = 0.02$) metastases per mouse, respectively. *Nalcn* deletion from $V1^{KP}$-IACs increased metastasis in particular to the peritoneum, kidneys and liver: *Nalcn* deletion from $Pdx1^{KP}$-PACs increased metastasis to the peritoneum and lungs (Fig. 4d). *Nalcn* deletion also increased metastasis of IAC and GAC in $P1^{KP}$ mice ($n = 80$) from $11.60 \pm 3.45$ metastases per $P1^{KP};Nalcn^{+/+}$ mouse to $42.21 \pm 11.23$ metastases per $P1^{KP};Nalcn^{+/Flx}$ mouse and $40.24.0 \pm 15.51$ metastases per $P1^{KP};Nalcn^{Flx/Flx}$ mouse (Fig. 4d and Supplementary Tables 9 and 12).

To further validate *Nalcn* loss-of-function as a driver of cancer metastasis, we treated additional cohorts of $V1^{KP};Nalcn^{+/+}$ ($n = 37$), $V1^{KP};Nalcn^{+/Flx}$ ($n = 17$) and $V1^{KP};Nalcn^{Flx/Flx}$ ($n = 8$) mice with $GdCl_3$ (2 μg per kg (body weight) per week). IACs in $GdCl_3$-treated $V1^{KP};Nalcn^{+/+}$ mice ($n = 28$) produced $18.32 \pm 5.95$ metastases per mouse compared with only $2.82 \pm 4.88$ in controls ($P = 0.02$; Fig. 4e and Supplementary Table 12). However, $GdCl_3$ did not increase metastasis in either $V1^{KP};Nalcn^{+/Flx}$ or $V1^{KP};Nalcn^{Flx/Flx}$ mice, confirming that the agent phenocopied the *Nalcn*-deletion metastatic phenotype, specifically.

**NALCN regulates CTCs.** Because *Nalcn* deletion increased tumor metastasis and the expression by GACs, IACs and PACs of genes enriched in human CTC transcriptomes (Fig. 3j), we reasoned that *Nalcn* might regulate the release of CTCs from primary tumors: CTCs are shed from tumors into the blood as precursors of metastasis[30]. To test this, nucleated, GAC, IAC and PAC cells that had been genetically tagged by recombination of the *Rosa26^{ZSG}* lineage-tracing allele in the corresponding epithelium were isolated from whole blood and quantified using ZsGreen (ZSG)-fluorescence-activated cell sorting (FACS). Serial, peripheral blood samples taken from *Prom1^{CreERT2/LacZ}* ($n = 397$), *Villin-1^{CreERT2}* ($n = 162$) or *Pdx1^{Cre}* ($n = 40$) mice that carried the *Rosa26^{ZSG}* allele and various combinations of oncogenic and *Nalcn^{Flx}* alleles were analyzed (Supplementary Table 13). An average ($\pm$ s.e.m.) of $3.8 \times 10^3 \pm 0.9 \times 10^3$ circulating ZSG$^+$ cells (CZCs) per ml of blood ($0.066\% \pm 0.02\%$ total cells) were isolated from all mice after an average of $254 \pm 9.1$ d following Cre-recombination (Fig. 5a–c and Supplementary Table 13). Across all three Cre-lines, the number of CZCs was highly correlated with both the presence of a primary tumor (Fig. 5b) and the total number of metastases (multiple linear regression, $T = 10.43$, $P = 0.000043$; Supplementary Table 13), independent of mouse sex or age of induction. *Nalcn* deletion, or $GdCl_3$ treatment, significantly increased the level of CZCs in tumor-bearing $P1^{KP}$, $V1^{KP}$ and $Pdx1^{KP}$ mice (Fig. 5c). Because neither *Prom1^{CreERT2/LacZ}*, *Villin-1^{CreERT2}* nor *Pdx1^{Cre}*

**Fig. 2 | NALCN regulates $P1^{KP}$-GAC proliferation and morphology. a**, Current responses to voltage steps from $-80$ to $80$ mV before (upper) and after (middle) addition of 100 μM gadolinium to $P1^{KP}$-GAC control, *Nalcn^{shRNA}*- and *NALCN^{cDNA}*-transfected cells, and current density responses to voltage steps before and after 100 μM gadolinium treatment ($n = 5$ biological replicate cells; values are mean $\pm$ s.e.m.). *$P$ values at $+40$, $+60$ and $+80$ mV for control gastric cells are 0.039, 0.032 and 0.023, respectively. $P$ values at $+40$, $+60$ and $+80$ mV for *NALCN^{cDNA}* cells are 0.013, 0.013 and 0.003, respectively (paired $t$-test). **b**, NALCN-mediated voltage-clamp ion currents from control, *Nalcn^{shRNA}*- and *NALCN^{cDNA}*-transfected $P1^{KP}$-GAC cells, and NALCN leak current density (current/cell membrane capacitance, mean $\pm$ s.e.m.) in control, *Nalcn^{shRNA}*- or *NALCN^{cDNA}*-transfected cells ($n = 5$ cells each, $P$ values comparing peak current at $+80$ mV voltage step are 0.05 control versus *NALCN^{cDNA}* cells and 0.023 control versus *Nalcn^{shRNA}* cells; Holm–Sidak multiple comparison procedure). **c**, Impact of gadolinium treatment (10 μM, $n = 386$ organoids; 100 μM, $n = 264$), *Nalcn^{shRNA}* ($n = 611$) or *NALCN^{cDNA}* ($n = 1,472$) transfection on $P1^{KP}$-GAC organoid size normalized to average $P1^{KP}$-GAC control treated organoids ($P1^{KP}$ 0 μM, $n = 663$ organoids, $4.274 \pm 0.6238$ (s.e.m.); $P1^{KP}$ shRNA control $n = 651$ organoids, $15.26 \pm 2.406$ (s.e.m.); $P1^{KP}$ cDNA control $n = 583$ organoids, $37.65 \pm 5.872$ (s.e.m.)). ***Exact $P < 0.0001$, two-tailed Mann–Whitney $U$-test. **d**, Representative macroscopic and photomicroscopic images of control ($n = 27$), *Nalcn^{shRNA}* ($n = 21$) or *NALCN^{cDNA}* ($n = 22$) transduced $P1^{KP}$-GAC orthotopic allografts. *Nalcn* RNA expression (in situ hybridization), and stromal (vimentin) and epithelial (CKAE1/AE3, CK7, CK5) marker immunohistochemistry are shown. Scale bar, 100 μm. **e**, *Nalcn* mRNA transcripts per tumor cell recorded in 20 individual tumor sections per treatment type. Bar, median. **$P = 0.0024$, ****$P = 0.00006$, two-tailed Mann–Whitney $U$-test.

recombine hematopoeitic cells in the bone marrow (Fig. 5d), these data strongly suggest that CZCs are CTCs shed from primary tumors through a process regulated by NALCN. In the immediate 5-week period following tamoxifen recombination, similar levels of circulating CZCs were observed among $P1^{KP}$ and $V1^{KP}$ mice that were $Nalcn^{+/+}$, $Nalcn^{+/Flx}$ or $Nalcn^{Flx/Flx}$, suggesting that Nalcn regulates

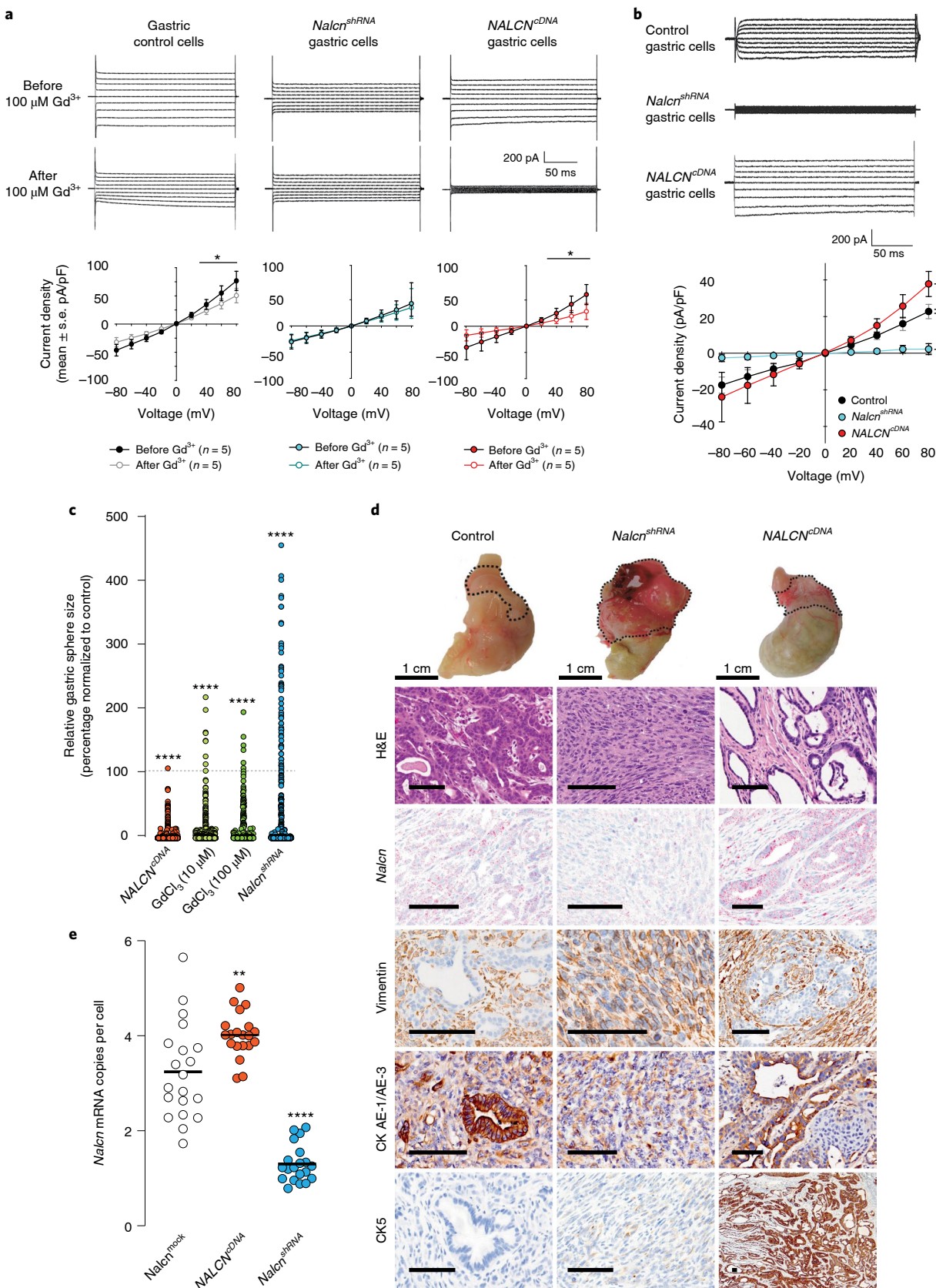

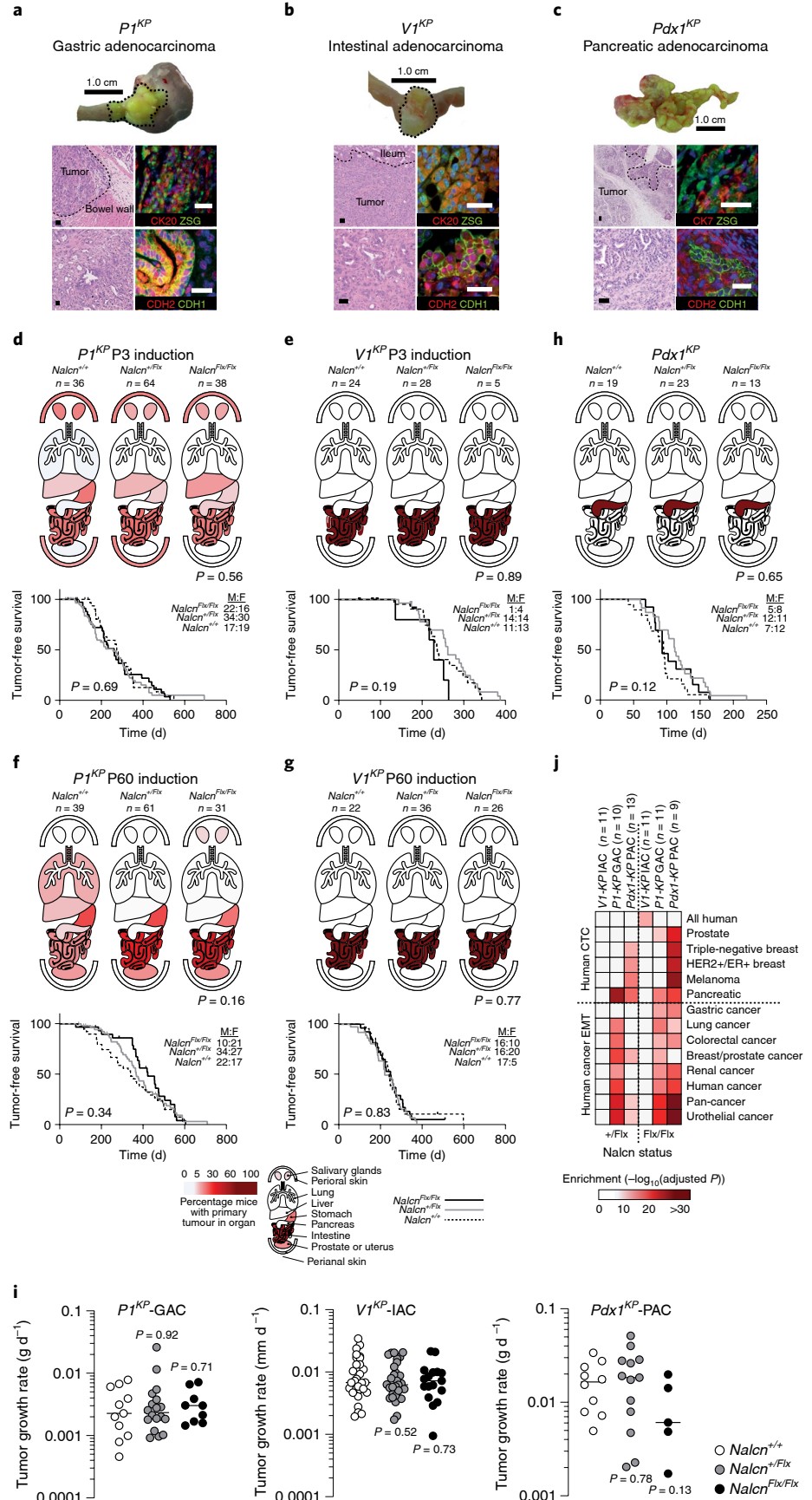

◄

**Fig. 3 | *Nalcn* deletion does not impact the incidence, tumor-free survival or growth rates of *P1^KP^*, *V1^KP^* or *Pdx1^KP^* primary tumors. a–c** Tumors and representative photomicrographs (H&E from all tumors (left; Supplementary Table 9) and dual immunofluorescence from five independent tumors each (right)) for lineage tracing (ZSG), epithelial (CK7, CK20) and EMT markers (CDH2, CDH1) of *P1^KP^*-GAC (**a**), *V1^KP^*-IAC (**b**) and *Pdx1^KP^*-PAC (**c**). Scale bar, 50 μm. Single-channel images are shown in Supplementary Fig. 1. **d–g**, Upper: organ heatmaps of tumor incidence in *P1^KP^* at P3 and *V1^KP^* at mice of each *Nalcn* genotype recombined at P3 (**d,e**) or P60 (**f,g**). Lower: survival curves of mice in each cohort. Male to female ration (M:F) is shown. *P1^KP^* P3, *P*=0.6912; *P1^KP^* P60, *P*=0.3897; *V1^KP^* P3, *P*=0.1900; and *V1^KP^* P60, *P*=0.8301. Mantel–Cox test. **h**, Organ primary tumor heatmaps and survival curves of *Pdx1^KP^* mice (*P*=0.1095). Mantel–Cox test. Source data for **d–h** are given in Supplementary Table 9. **i**, Growth rates of *P1^KP^*-GAC (*n*=38), *V1^KP^*-IAC (*n*=57) and *Pdx1^KP^*-PAC (*n*=28) tumors. Two-tailed Mann–Whitney *U*-tests revealed no significant difference in growth rates among tumors with different *Nalcn* genotypes *P1^KP^*-GAC: *Nalcn^+/+^* (*n*=11) versus *Nalcn^+/Flx^* (*n*=18; *P*=0.912), versus *Nalcn^Flx/Flx^* (*n*=9; *P*=0.7103). *V1^KP^*-IAC: *Nalcn^+/+^* (*n*=16) versus *Nalcn^+/Flx^* (*n*=25; *P*=0.5169), versus *Nalcn^Flx/Flx^* (*n*=16; *P*=0.7309). *Pdx1^KP^*-PAC: *Nalcn^+/+^* (*n*=10) versus *Nalcn^+/Flx^* (*n*=13; *P*=0.7844), versus *Nalcn^Flx/Flx^* (*n*=5; *P*=0.1292). Bar, median. Source data are given in Supplementary Table 10. **j**, Gene set enrichment analyses of transcriptomes of *Nalcn^+/Flx^* and *Nalcn^Flx/Flx^* *P1^KP^*-GAC, *V1^KP^*-IAC and *Pdx1^KP^*-PAC versus *Nalcn^+/+^* tumors.

cell shedding as a late event (Fig. 5e,f and Supplementary Table 13); however, the time taken for lineage tracing to reach steady state in our mice may underestimate CZC numbers at early time points.

To better understand the origin of CZCs, we generated single-cell RNA sequence profiles of CZCs isolated from mice with *P1^KP^*-GAC (*n*=1,701 cells) or *V1^KP^*-IAC (*n*=119), as well as peripheral blood mononuclear cells (PBMCs, *n*=559; Fig. 6a), and compared these with published single-cell RNA sequence profiles of human breast, lung, pancreatic and prostate CTCs (*n*=360) and PBMCs (*n*=500)[31–36]. Human CTCs comprised three overlapping clusters (Extended Data Fig. 5a–c and Supplementary Tables 14 and 15): 'huCTC1' (enriched with cancer metastasis, EMT and epithelial gene sets); huCTC3 (enriched with early-erythroid and EMT gene sets); and huCTC2 (sharing profiles of huCTC1 and huCTC3). huCTC1–3 expressed β-globin (*HBB*)—a survival factor for human CTCs[33]—as well as *HBA1*, *HBA2* and *HBD*. Mouse CZCs formed seven clusters whose transcriptomes significantly matched huCTC1 (mCZC2–7), huCTC2 (mCZC2, 3, 5–7) and huCTC3 (mCZC2–7), and included orthologs of *HBA1*, *HBA2* (*Hba-a1*, *Hba-a2*), *HBB* (*Hbb-bs*, *Hbb-bt*), *ANXA2* and *LGALS3*, as well as genes expressed in normal and malignant stomach and small intestine (Fig. 6a,b, Extended Data Fig. 5c–g and Supplementary Tables 16 and 17). Normalization and Uniform Manifold Approximation and Projections (UMAP) of all single-cell RNA sequence profiles also revealed significant overlap in mouse CZC and human CTC transcriptomes, especially those enriched for CD71+ erythroid genes (Extended Data Fig. 5e–g and Supplementary Table 18). Coimmunofluorescence of peripheral blood smears taken from mice with *V1^KP^*-IAC and *P1^KP^*-GAC confirmed CZC expression of HBA-A1, LGALS3, and epithelial cell markers (KRT80, CDH1) and CDX2 that marks intestinal epithelium (Fig. 6c). PBMCs did not express these markers but did express markers of PBMCs (for example, CD45).

To test directly whether CZCs possess metastatic potential, we injected separate aliquots of 25,000 CZCs isolated from mice with *P1^KP^*-PAC, *P1^KP^*-GAC or *V1^KP^*-IAC into the tail veins of

immunocompromised mice. Within 75 d, all mice developed numerous ZSG+ metastases in the lungs, liver, kidneys and/or peritoneum (Fig. 6d,e and Supplementary Table 19). Similar studies with increasing cell dilutions showed that as few as ten CZCs were required to generate metastasis (Fig. 6f and Supplementary Table 19). Thus, CZCs are highly enriched for CTCs that recapitulate the transcriptome of human CTCs and are shed into the peripheral blood through a process regulated by *Nalcn*.

**NALCN and circulating noncancer cells.** Preventing CTC shedding into the peripheral blood could stop metastasis, but disentangling this process from the complex cascade of tumorigenesis has proved challenging. Deletion of *Nalcn* from freshly isolated *P1;Nalcn^Flx/Flx^* gastric stem cells that lacked oncogenic alleles, rapidly upregulated genes associated with invasion (for example, *Mmp7*, *Mmp9*, *Mmp10* and *Mmp19*) and gastric EMT (for example, *Zeb1*, *Fstl1*, *Sparc*, *Sfrp4*, *Cdh6* and *Timp3*; Supplementary Tables 20 and 21), suggesting NALCN might regulate cell shedding from solid tissues independent of transformation. To test this, we looked for CZCs in the peripheral blood of *Prom1^CreERT2/LacZ^*; *Rosa26^ZSG^;Nalcn^+/+^* (*P1^R^Nalcn^+/+^*, *n*=87), *P1^R^Nalcn^+/Flx^* (*n*=50) and *P1^R^Nalcn^Flx/Flx^* (*n*=37) mice that lacked oncogenic alleles and never developed tumors (Supplementary Table 13). Remarkably, deletion of *Nalcn* increased the numbers of CZCs in these mice to levels similar to those observed in tumor-bearing animals (Figs. 5b,c and 7a). Single-cell RNA sequencing (SCS) profiles of CZCs isolated from nontumor-bearing (ntCZC) mice co-clustered with CZCs from tumor-bearing animals (tCZC; Fig. 7b). The great majority of tCZC and ntCZC SCSs did not cluster with SCS profiles of primary IACs, GACs or normal tissues, but with SCS profiles of metastases (Fig. 7b and Supplementary Table 22). SCS profiles of both tCZCs and ntCZCs matched those of human CTCs and, similar to human CTCs[2], expressed genes associated with stem and progenitor cells; although tCZCs were relatively more enriched for metastasis and invasion-associated gene sets (Extended Data Fig. 6a and Supplementary Tables 23 and 24).

**Fig. 4 | NALCN loss-of-function increases tumor metastasis. a**, Unsupervised hierarchical clustering of *P1^KP^* (GAC, *n*=10; lung adenocarcinoma, *n*=6; prostatic adenocarcinoma, *n*=2), *V1^KP^* (IAC, *n*=19), *Pdx1^KP^* (PAC, *n*=13) and *P1;Pten^Flx/Flx^;Trp53^Flx/Flx^* (*P1^PtP^*) (hepatobiliary, *n*=3; lung adenocarcinoma, *n*=1) primary tumors and metastatic (liver, *n*=2; peritoneum, *n*=11; kidney, *n*=1; thoracic cavity, *n*=4; lung, *n*=1; lymph node, *n*=2) tumors. Heatmap reports enrichment of primary tumor transcriptomes in metastatic tumors. **b**, Exemplar ZSG+ metastatic tumors (met, outlined). Scale bar, 0.5 cm. **c**, Photomicrographs (H&E (left) and immunohistochemistry/fluorescence(right)) of the metastases in **b**. Scale bar, 50 μm. All enumerated metastases were evaluated using H&E (full list is given in Supplementary Table 9; *n*=7,076 metastases); *n*=59 metastases were evaluated by ZSG for IHC and *n*=20 metastases were evaluated by immunofluorescence. Single-channel images are shown in Supplementary Fig. 1. **d**, Left: cumulative total number of adenocarcinoma metastases per mouse post Cre-recombination (two-tailed Mann–Whitney *U*-test, total tumor burden in *Nalcn*-deleted versus wild-type mice; Supplementary Table 9). Right: total metastases per mouse in anatomical regions. Male/female (M:F) and P3/P60 mice are shown. *V1^KP^* IAC for individual organs: liver, *P*=0.0371 (*Nalcn^Flx/Flx^*); kidney, *P*=0.0229 (*Nalcn^Flx/Flx^*); and peritoneum, *P*=0.0492 (*Nalcn^+/Flx^*) and **P*=0.0015 (*Nalcn^Flx/Flx^*). *Pdx1^KP^* PAC individual organs: lung, *P*=0.0328 (*Nalcn^+/Flx^*); and peritoneum, **P*=0.0050 (*Nalcn^+/Flx^*). *P1^KP^* GAC and IAC individual organs: lung, **P*=0.0085 (*Nalcn^+/Flx^*) and **P*=0.0048 (*Nalcn^Flx/Flx^*). **e**, Metastatic burden and organ metastases in *V1^KP^*-IAC gadolinium or control treated mice. **P*=0.0090, two-tailed Mann–Whitney *U*-test.

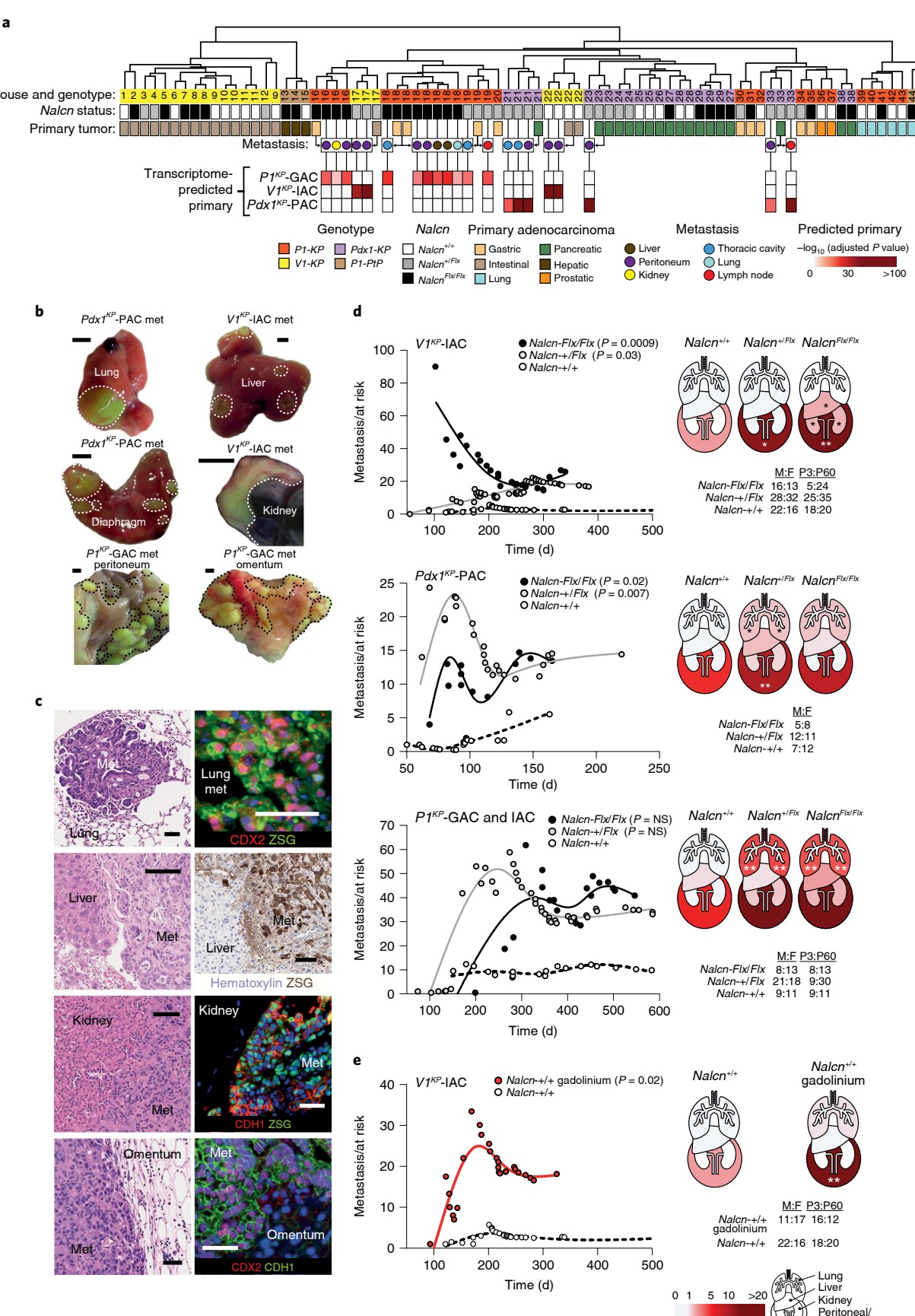

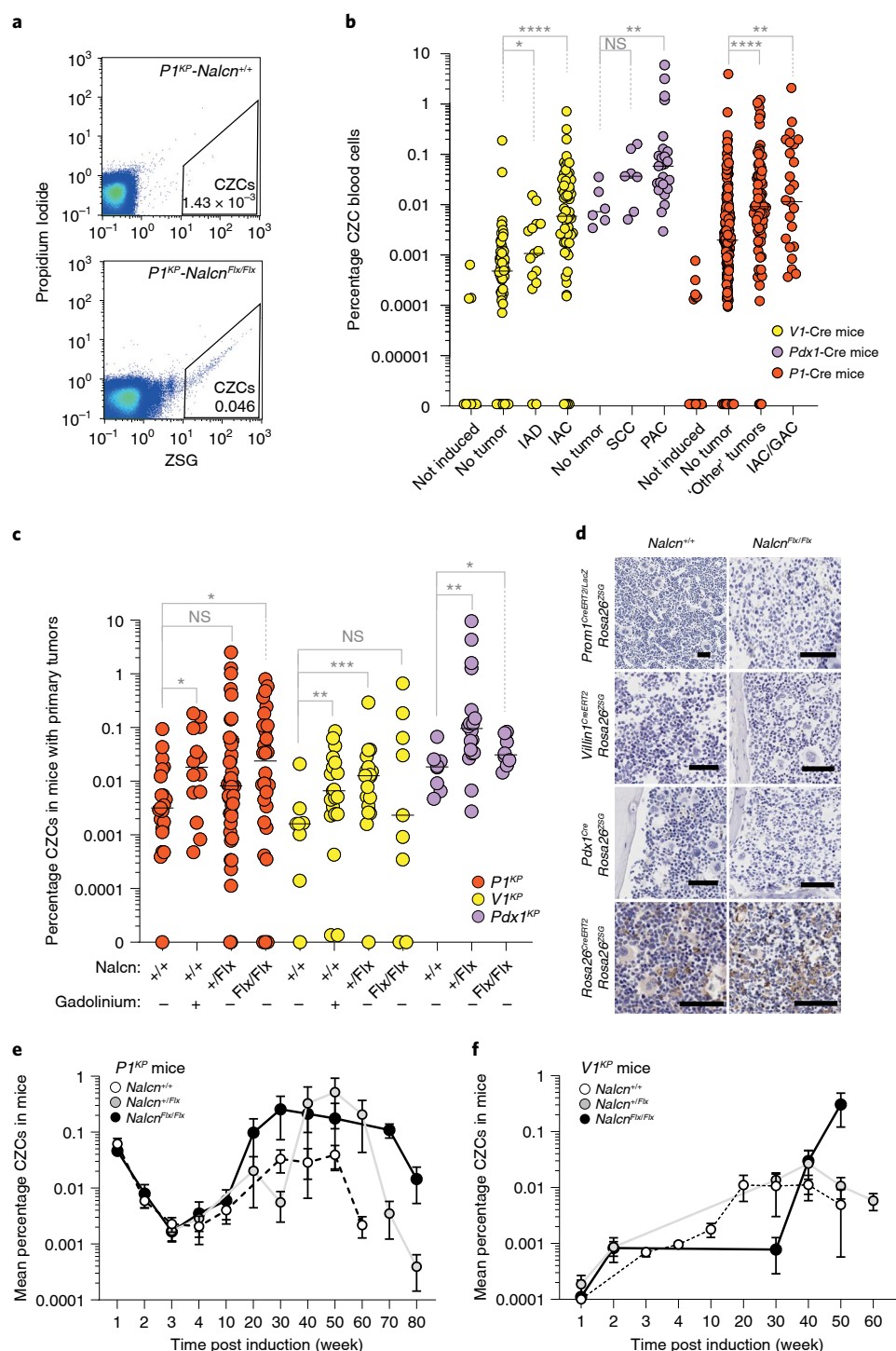

**Fig. 5 | NALCN loss-of-function increases nucleated CZCs in *P1^KP^, V1^KP^* and *Pdx1^KP^* mice. a**, FACS profiles gating CZCs in blood samples of *P1^KP^ Nalcn^+/+^* and *Nalcn^Flx/Flx^* mice (per cent nucleated cells). Gating strategy is shown in Supplementary Fig. 2. **b**, Scatter plot of CZCs (per cent of total nucleated blood cells) of *Prom1^CreERT2/LacZ^* (*n* = 397), *Villin-1^CreERT2^* (*n* = 162) or *Pdx1^Cre^* (*n* = 40) mice that did, or did not, contain a primary tumor. Data are biologically independent peripheral blood samples. Bar, median. V1-Cre: *\*P* = 0.0499, \*\*\*\**P* < 0.0001; Pdx1-Cre: not significant (NS) *P* = 0.0513, \*\**P* = 0.0033; P1-Cre: \*\**P* = 0.0033, \*\*\*\**P* < 0.0001; two-tailed Mann–Whitney *U*-test. Source data are available in Supplementary Table 13. **c**, Scatter plot of CZCs according to genotype and gadolinium treatment in tumor-bearing animals. Data are biologically independent peripheral blood samples. Bar, median. *P1^KP^* (*n* = 112): *\*P* = 0.02, NS *P* = 0.1204; *V1^KP^* (*n* = 64): \*\**P* = 0.0088, \*\*\**P* = 0.0004, NS *P* = 0.4213; *Pdx1^KP^* (*n* = 34): *\*P* = 0.0499, \*\**P* = 0.0027; two-tailed Mann–Whitney *U*-test. Source data are available in Supplementary Table 13. **d**, Representative photomicrographs of ZSG immunohistochemistry of bone marrow of mice of the indicated genotype at a minimum of 100 d post Cre-recombination. Scale, 100 μm. Three mice were evaluated for each Cre strain. **e,f**, FACS quantification of CZCs in *P1^KP^* (*Nalcn^+/+^, n* = 11; *Nalcn^+/Flx^, n* = 4; *Nalcn^Flx/Flx^, n* = 6) (**e**) and *V1^KP^* (*Nalcn^+/+^, n* = 9; *Nalcn^+/Flx^, n* = 4; *Nalcn^Flx/Flx^, n* = 4) (**f**) mice (mean ± s.e.m.) from 1-week post tamoxifen induction. Source data are given in Supplementary Table 13.

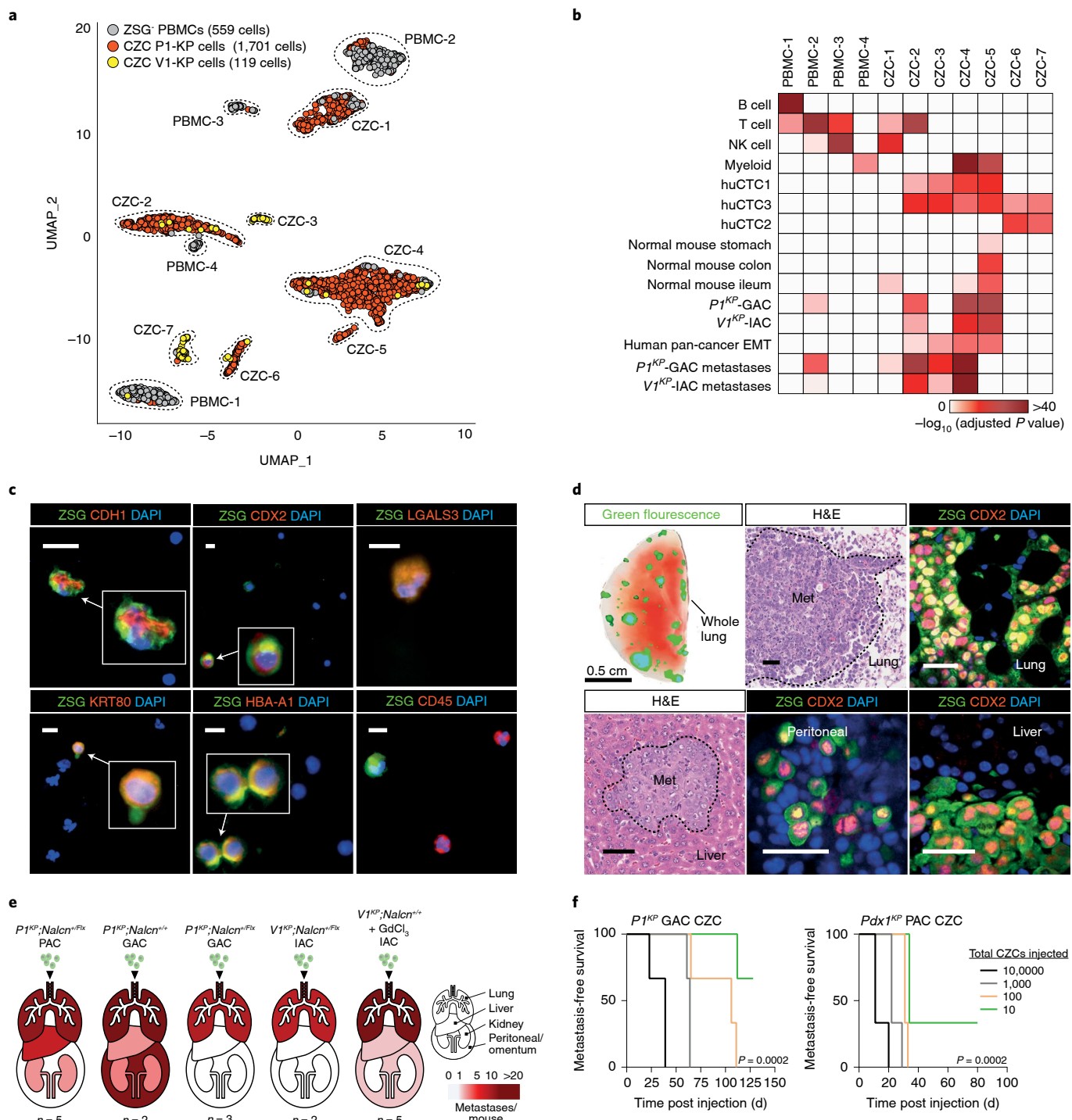

**Fig. 6 | Nucleated CZCs in *P1^{KP}*, *V1^{KP}* and *Pdx1^{KP}* mice are CTCs. a**, UMAP of SCS profiles of CZCs (*n* = 1,820) and PBMCs (*n* = 559). **b**, Gene set enrichment from 2,086 gene sets in UMAP clusters in **a**. **c**, Coimmunofluorescence of CZCs and PBMCs in *P1^{KP}* (upper) and V1^{KP} (lower) mice (ZSG; scale bar, 10 μm). Representative photomicrographs of 22 cells identified across *n* = 20 blood films assessed from *n* = 5 tumor-bearing animals. **d**, Autofluorescence of *Pdx1^{KP}*-PAC CZC metastases in whole lung of recipient immunocompromised mouse (upper left; scale bar, 0.5 cm). Other images show H&E (representative image of 3,061 metastases evaluated) or coimmunofluorescence of metastases (representative images of 28 metastases evaluated) of *P1^{KP}*-GAC or *V1^{KP}*-IAC CZC metastases in recipient mice (scale bar, 50 μm). Single-channel images are shown in Supplementary Fig. 1. **e**, Total metastases per organ in recipient mice injected with 25,000 CZCs. P1^{KP} *Nalcn^{+/Flx}* PAC, *n* = 5 mice; P1^{KP} *Nalcn^{+/+}* GAC, *n* = 2 mice; P1^{KP} *Nalcn^{+/Flx}* GAC, *n* = 3 mice; V1^{KP} *Nalcn^{+/Flx}* IAC, *n* = 2 mice; V1^{KP} *Nalcn^{+/+}* + GdCl₃ IAC, *n* = 5 mice. Source data are given in Supplementary Table 19. **f**, Metastasis-free survival of immunodeficient NOD scid gamma recipient mice injected with different numbers (10,000, 1,000, 100 or 10) of *P1^{KP}* GAC or *Pdx1^{KP}* PAC CZCs (*n* = 3 mice for each condition). \*\*\**P* = 0.0002 Mantel–Cox statistic. Source data are available in Supplementary Table 19.

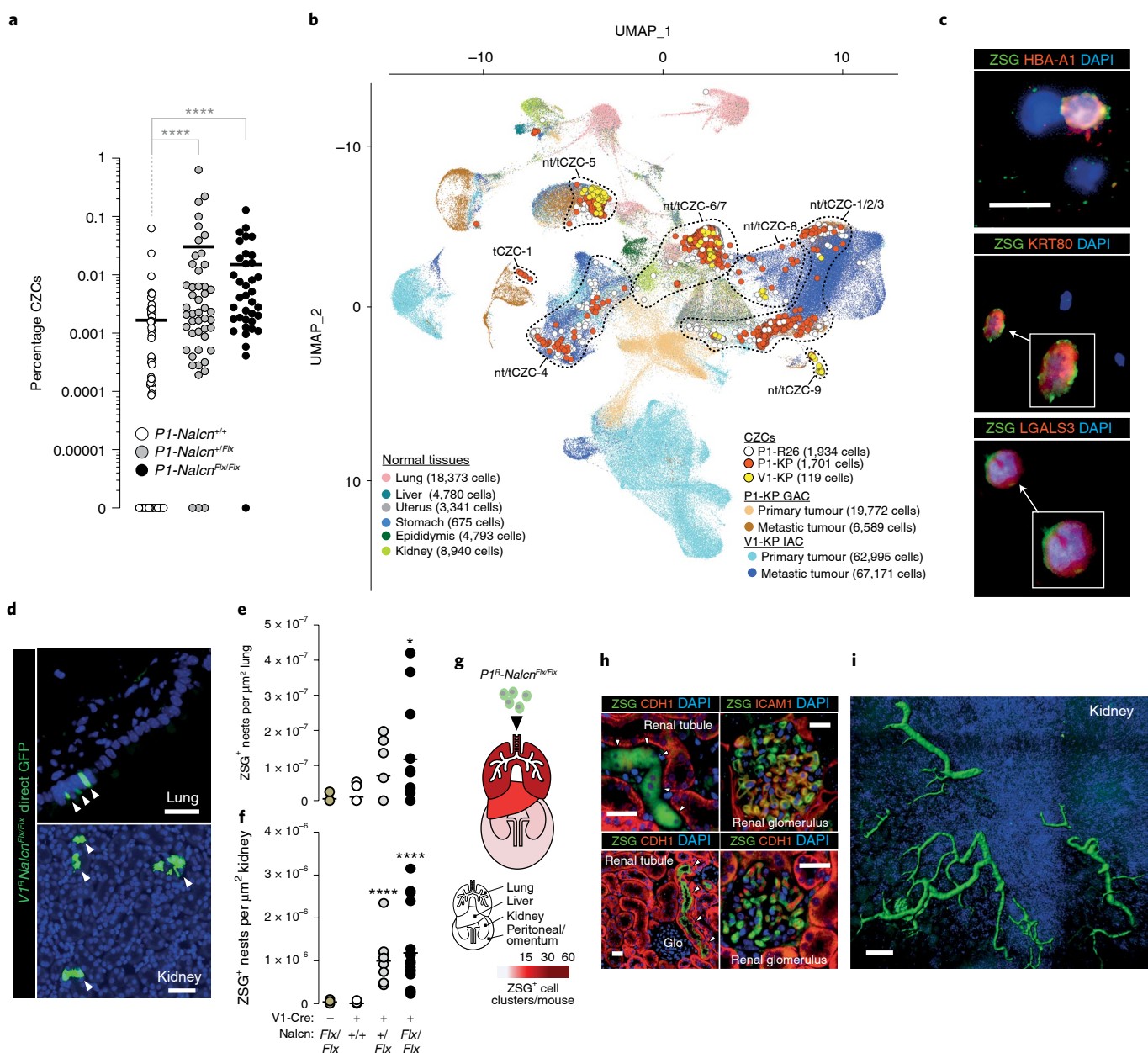

**Fig. 7 | NALCN loss-of-function increases shedding of ntCZCs. a**, ntCZCs identified in individual nontumor-bearing *P1^RNalcn^+/+* (*n* = 87), *P1^RNalcn^+/Flx* (*n* = 50) and *P1^RNalcn^Flx/Flx* (*n* = 37) mice. Bar, median. \*\*\*\**P* < 0.0001, two-tailed Mann–Whitney *U*-test. Source data are available in Supplementary Table 13. **b**, UMAP of 201,183 SCS profiles of PBMCs, tCZCs and ntCZCs as well as cells derived from the indicated normal and malignant mouse tissues. **c**, Coimmunofluorescence of ntCZCs and PBMCs in peripheral blood smears of *P1^RNalcn^Flx/Flx* mice (ZSG; scale bar, 10 μm). Representative photomicrographs of 11 cells identified in *n* = 20 blood films from *n* = 4 mice. Single-channel images are shown in Supplementary Fig. 1. **d–f**, Direct ZSG-immunofluorescence photomicrographs of ZSG+ cells in lung and kidney (scale bar, 50 μm) (**d**), and enumerated in lung (no Cre, *n* = 2 mice, 5 lung lobes; *Nalcn*+/+, *n* = 3 mice, 9 lung lobes; *Nalcn*+/Flx, *n* = 3 mice, 8 lung lobes; *Nalcn*Flx/Flx, *n* = 5 mice, 12 lung lobes; NS *P* = 0.1312, \**P* = 0.0168, two-tailed Mann–Whitney *U*-test) (**e**) and kidney (no Cre, *n* = 2 mice, 4 kidney sections; *Nalcn*+/+, *n* = 3 mice, 11 kidney sections; *Nalcn*+/Flx, *n* = 3 mice, 10 kidney sections; *Nalcn*Flx/Flx, *n* = 5 mice, 18 kidney sections; \*\*\*\**P* < 0.0001, two-tailed Mann–Whitney *U*-test) (**f**). **g**, Organ heatmap of total numbers of ZSG+ cell clusters per mouse identified in organs of recipient mice injected with *P1^RNalcn^Flx/Flx* ntCZCs. **h**, Coimmunofluorescence of *P1^RNalcn^Flx/Flx* ntCZCs (arrows) incorporated into the kidneys of recipient mice (arrows indicated ZSG+ cells; scale bar, 50 μm). Representative photomicrograph of *n* = 5 ZSG rests identified in one tissue field from *n* = 5 mice. Single-channel images are shown in Supplementary Fig. 1. GLO, glomerulus. **i**, Confocal laser scanning microscope image of *P1^RNalcn^Flx/Flx* CZCs incorporated into the renal cortex of recipient mice. Scale bar, 100 μm. Representative image of *n* = 2 mouse kidneys assessed.

Coimmunofluorescence of blood smears confirmed that both ntCZCs and tCZCs share markers of huCTCs, including HBA-A1 (Figs. 6c and 7c).

To understand the fate of ntCZCs, we looked for ZSG+ cells in the lungs and kidneys of aged *V1^R* and *Pdx1^R Nalcn*+/+, *Nalcn*+/Flx and/or *Nalcn*Flx/Flx mice. Remarkably, ZSG+ cell clusters were readily detected in these organs in *Nalcn*-deleted animals, but were absent or detected at significantly lower levels in *Nalcn*+/+ mice, suggesting that ntCZCs traffic to, and embed within, distant organs (Fig. 7d–f and Extended Data Fig. 6b,c). To test this more directly, we injected

separate aliquots of 25,000 ntCZCs isolated from $P1^RNalcn^{Flx/Flx}$ mice into the tail veins of six immunocompromised mice. All recipient mice remained clinically well after an average of 100 d, but contained numerous ZSG$^+$/Cdh1$^+$/Icam1$^+$ donor-cell clusters within their lungs, liver, kidneys and peritoneum at a frequency similar to metastatic tumors formed by tail-vein injections of tCZCs (Figs. 6e, 7g–i and Extended Data Fig. 6d). Trafficked ntCZCs formed apparently normal structures in target organs, the most extreme example being kidney glomeruli and tubules (Fig. 7h,i). Thus, NALCN regulates cell shedding from solid tissues independent of cancer, divorcing this process from tumorigenesis and unmasking an oncogene-independent metastatic pathway.

**NALCN-blockade causes systemic fibrosis.** Although $P1^RNalcn^{+/Flx}$ ($n = 118$) and $P1^RNalcn^{Flx/Flx}$ ($n = 112$) mice did not develop cancer, whole-body autopsy of these mice revealed severe kidney and skin fibrosis relative to $P1^RNalcn^{+/+}$ ($n = 65$) mice (Supplementary Table 25 and Extended Data Fig. 7). This pathology arose after ≥400 d and replicated that of gadolinium-induced systemic fibrosis (GISF), a debilitating condition manifested by severe organ fibrosis following administration of gadolinium-based contrast agents[37]. How gadolinium-based contrast agents cause GISF is unknown, but suggested mechanisms include tissue retention of gadolinium-based contrast agents and the mobilization and recruitment of bone marrow-derived fibrocytes[38]. Our data suggest strongly that blockade of the NALCN channel by gadolinium mobilizes epithelial cells in a variety of epithelial tissues that traffic to the kidney and other organs, eventually eliciting a fibrotic response, causing GISF.

## Discussion

Developing antimetastatic therapies has proven difficult because targets in primary tumors that drive metastasis have proved hard to find[2]. By divorcing the process of CTC shedding from 'upstream' tumorigenesis, our data unmask manipulation of NALCN function as a promising new approach to block metastasis. In particular, drugs capable of reopening the NALCN channel might be effective antimetastatic therapies. Precedent for this approach is provided by drugs that open the chloride channel mutated in cystic fibrosis[39]. If successful, such agents may also be useful for treating GISF.

It is important to note that our observations are based on deleting *Nalcn* from mouse tissues, whereas *NALCN* in human cancers is affected predominantly by nonsynonymous mutations. Although our in silico modeling suggests strongly that these cancer-associated mutations close the *NALCN* channel, it will be important to demonstrate this functionally by modeling nonsynonymous *Nalcn* mutations in vivo. These studies should also include testing in patient-derived xenografts of gastric, colon and other cancers to confirm that NALCN regulates trafficking of human as well as mouse cells.

Loss-of-function mutations in *NALCN* may also help explain various enigmatic features of human cancer. Metastases can emerge many years after removal of a localized cancer[40], or in the absence of a primary tumor[41]. Loss of NALCN function in our mice caused an abundant and persistent shedding of cells that embed in distant organs, even in the absence of a primary tumor. Because human epithelial tissues contain fields of phenotypically normal cells that harbor oncogenic mutations[42,43], then loss of NALCN function in these cells could provide a source of CTCs that form metastases in the absence of a primary tumor, or long after a primary tumor has been removed. It is likely that such cells would need to acquire additional mutations to form tumors at the metastatic site, compatible with the relative rarity of these phenomena. Our data may also explain why CTCs have been found in the bone marrow of patients who lack metastases. Although these cells could represent 'dormant' CTCs, as previously suggested[3], equivalent to ntCZCs in our mice, they may be shed from nontransformed epithelia that have lost NALCN

function, but not gained the ability to form metastatic tumors. Our serial analysis of CZCs in mice suggest that cell shedding following NALCN loss-of-function is a late, rather than early, event; although *NALCN* mutations could promote both linear and parallel progression models of cancer[44].

Our data also provide clues as to how NALCN might regulate epithelial cell shedding. We observed upregulation of genes associated with EMT and invasion within 72 h of deleting *Nalcn* from normal gastric stem cells; suggesting that this channel might regulate gene transcription in a similar manner to that reported for calcium ion channels[6,45]. Our electrophysiology studies demonstrate that GAC cells possess a NALCN-mediated current. However, more detailed electrophysiology studies are required to determine the precise mechanism by which NALCN regulates gene expression and cell shedding and whether this involves maintenance of the resting membrane potential.

The development of renal and skin fibrosis reminiscent of GISF in aged *Nalcn*-deleted mice, pinpoint NALCN channel blockade as the likely cause of this debilitating condition. $P1^{KP}$ mice succumbed to cancer well before the onset of organ fibrosis in $P1^R$ mice, and *Nalcn* deletion in $P1^R$ mice did not induce stomach, intestine, lung, pancreas or liver fibrosis—principal sites of primary and metastatic tumors in $P1^{KP}$ mice. Thus, fibrosis is unlikely to have contributed to metastasis in *Nalcn*-deleted mice. However, because limited exposure to gadolinium can induce GISF in humans, it is a note of concern that gadolinium-contrast imaging of cancer patients could accelerate metastasis.

## Online content

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

## Methods

**Culture of stomach stem cells.** Gastric glands were isolated[46] by perfusing mice with 30 mM EDTA/PBS, stomach removal and scraping pyloric mucosa into 10 mM EDTA/PBS at 4 °C. Dissociated, filtered and resuspended cells were placed in Matrigel (catalog number 354230, BD Biosciences) and culture medium: advanced DMEM/F12 (catalog number 31330038, Thermo Fisher Scientific), B27 (catalog number 12587010, Thermo Fisher Scientific), N2 (catalog number A1370701, Thermo Fisher Scientific), N-acetylcysteine (catalog number A9165, Sigma-Aldrich) and 10 nM gastrin (catalog number G9145, Sigma-Aldrich) containing growth factors (50 ng ml$^{-1}$ EGF (PeproTech), 1 mg ml$^{-1}$ R-spondin1 (catalog number 120-38, PeproTech), 100 ng ml$^{-1}$ Noggin (catalog number 250-38, PeproTech), 100 ng ml$^{-1}$ FGF10 (catalog number 100-26, PeproTech) and Wnt3A conditioned media (L Wnt-3A, catalog number ATCC-CRL-2647, American Type Culture Collection). Gastric spheres were passaged by dispase (catalog number D4818, Sigma-Aldrich) digestion and dissociation into single cells (StemPro Accutase, Life Technologies). Gadolinium (catalog number 439770, Sigma-Aldrich) was diluted in the culture medium and overlaid on Matrigel embedded cells (Supplementary Tables 26 and 27).

**Lentiviral production and transduction.** Nalcn-shRNA lentivirus was produced as described previously[47]. Three shRNAs per target (two open reading frames one 3′-untranslated region) were cloned into pFUGWH1-RFPTurbo and cotransfected with plasmids pVSV-G and pCMVd8.9 into 293FT (Thermo Fisher Scientific, catalog number R70007) cells. NALCN cDNA (NM_052867) was from OriGene (catalog number RC217074). In total $2 \times 10^4$ gastric cells were mixed with lentiviruses (20 particles per cell) plated in Matrigel. Transduced red fluorescent$^+$ (shRNA) or green fluorescent$^+$ (cDNA) cells were sorted using a Becton Dickinson Aria II Cell Sorter (Supplementary Tables 26 and 28).

**Whole-cell electrophysiology.** The NALCN channel current was measured as reported[48]. Whole-cell recordings were obtained from stomach tumor cells on 12-mm cover slips coated with Matrigel at a density of 25,000 cells per ml and superfused (2–3 ml min$^{-1}$) with warm (30–32 °C) recording solution containing 120 mM NaCl, 5 mM CsCl, 2.5 mM KCl, 2 mM CaCl$_2$, 2 mM MgCl$_2$, 1.25 mM NaH$_2$PO$_4$, 26 mM NaHCO$_3$, 20 mM glucose and 1 M tetrodotoxin (300–310 mOsm), with 95% O$_2$/5% CO$_2$. Patch pipettes (open pipette resistance, 3–4 MΩ) were filled with an internal solution containing 125 mM CsMeSO$_3$, 2 mM CsCl, 10 mM HEPES, 0.1 mM EGTA, 4 mM MgATP, 0.3 mM NaGTP, 10 mM Na$_2$ creatine phosphate, 5 mM QX-314 and 5 mM tetraethylammonium Cl (pH 7.4, adjusted with CsOH, 290–295 mOsm). Tetrodotoxin and QX-314 were included to block voltage-sensitive sodium channels in recorded cells, whereas cesium and tetraethylammonium Cl blocked voltage-sensitive potassium channels. Voltage-clamp recordings were made using a Multiclamp 700B (Molecular Devices), digitized (10 kHz; DigiData 1322A, Molecular Devices) and recorded using pCLAMP v.10.0 software (Molecular Devices). In all experiments, membrane potentials were corrected for a liquid junction potential of −10 mV. After forming a gigaseal onto a cell and rupturing the cell membrane, tumor cell membrane potential was held at −70 mV. Cell membrane capacitance, membrane resistance and pipette access resistance were then measured with the pCLAMP cell membrane test function. Recordings were excluded if pipette access resistance was higher than 20 MΩ or if access resistance changed by more than 20% during the experiment. After cell membrane resistance had stabilized, membrane potential was then stepped to 0 mV for 100 ms followed by a series of 250 ms voltage steps from −80 mV to +80 mV in 20-mV increments and the current response to these voltage steps was recorded. GdCl$_3$ (100 μM) was then applied to the bath solution to eliminate the voltage-independent 'leak' current associated with Nalcn. Calculation of the Nalcn current was performed offline by subtracting the current response in GdCl$_3$ from the previous GdCl$_3$-free current recording. Tumor cell Nalcn current density was determined by dividing the Nalcn current by cell membrane capacitance. To verify successful expression of the RFP$^+$ (Nalcn$^{shRNA}$) or GFP$^+$ (NALCN$^{cDNA}$) construct, cells were imaged with two-photon laser scanning microscopy (Prairie Technologies) using a Ti:sapphire Chameleon Ultra femtosecond-pulsed laser (Coherent), and ×60 (0.9 NA) water-immersion infrared objective (Olympus). Red fluorescent protein was visualized using an excitation wavelength of 1030 nM, whereas green fluorescent protein (GFP) was visualized using an excitation wavelength of 820 nM (Supplementary Tables 26 and 28).

**Gastric adenocarcinoma allografts.** $P1^{KP}$-GAC orthotopic and flank allografts were generated under protocols approved by the Institutional Animal Care and Use Committee of St. Jude Children's Research Hospital (IACUC-SJ). For orthotopic grafts, a longitudinal abdominal incision was made to expose the pyloric valve of CD-Foxn1$^{NU}$ mice and $2 \times 10^5$ freshly dissociated $P1^{KP}$-GAC cells suspended in Matrigel and fast green (Santa Cruz) were injected into the pyloric stomach epithelium. The wound was closed and mice were monitored daily for tumor development. Under veterinary guidance and IACUC-SJ approved measures, animals reaching humane end points were immediately euthanized and a full autopsy completed (Supplementary Tables 26 and 29).

**Generation of Nalcn$^{Flx}$ allele.** Mice were derived from targeted embryonic stem cells (ESCs) (UCDAVIS KOMP Repository Knockout Mouse Project clone EPD0383_5_C01). ESCs were screened using KOMP PCR strategies for Nalcntm1a(KOMP)Wstsi. ESCs were implanted into recipient C57/Bl6 mice in accordance with protocols approved by IACUC-SJ. Wild-type Nalcn and Nalcn$^{Flx}$ alleles were detected using standard PCR and primers (UCDAVIS KOMP Repository Knockout Mouse Project clone EPD0383_5_C01). Nalcn RNA expression was quantified by quantitative PCR (qPCR) with reverse transcription and a Bio-Rad CFX96 Touch Real-Time PCR Detection System with primers (see Supplementary Tables 26 and 29–31 for details on animals and oligonucleotide sequences).

**Tumorigenesis and surveillance.** All animal studies within the United Kingdom (UK) were performed under the Animals (Scientific Procedures) Act 1986 in accordance with UK Home Office licenses (Project License 70-8823, P47AE7E47, PP7834816) and approved by the Cancer Research UK (CRUK) Cambridge Institute Animal Welfare and Ethical Review Board. Mice were housed in individually ventilated cages with wood chip bedding and nestlets with environmental enrichment (cardboard fun tunnels and chew blocks) under a 12 h light/dark cycle at 21 ± 2 °C and 55% ± 10% humidity. Diet was irradiated LabDiet 5R58 with ad libitum water. Animals carrying the modified Nalcn allele were bred to RosaFLPe-expressing mice to remove LacZ and Neo cassette. Animals with complete recombination were bred with: Prom1C-L[29]; Nestin-cre[49]; Rosa-CreERT[50]; villin-CreER[25]; Pdx1-cre[28]; RosaZSG[51]; and KrasG12D/+[52], Trp53flx[53]. Cre-recombination was activated by dosing with 1 mg of tamoxifen per 40 g (body weight) at P3 or 8 mg tamoxifen per 40 g (body weight) at P60. Mice were maintained for up to 2 years and full-body autopsy was performed as described[4] at humane end points or the indicated time point, whichever was first. All tissues were inspected for macroscopic tumors with direct green fluorescence detection. Tissues were formalin fixed, paraffin embedded with portions also snap frozen or used for tissue dissociation for sequencing (Supplementary Tables 26 and 29).

**Histology.** Hematoxylin and eosin (H&E) staining was performed using standard procedures (catalog number 7221, 7111, Thermo Fisher Scientific). Fibrosis was assessed using modified Masson's trichrome and Picrosirius Red stains. Immunohistochemistry was performed using standard procedures and primary antibodies: Ki67 (catalog number IHC-00375, Bethyl Laboratories, 1:1,000), ZSG (catalog number 632474, Clontech, 1:2,000), pan cytokeratin (AE1/AE3) (catalog number 901-011-091620, BioCare Medical, 1:100), CK5 (catalog number ab52635, Abcam, 1:100), vimentin (catalog number 5741S, Cell Signaling Technology, 1:200), cleaved caspase 3 (catalog number 9664, Cell Signaling Technology, 1:200), CD31 (catalog number 77699, Cell Signaling Technology, 1:100), α-smooth muscle actin (catalog number ab5694, Abcam, 1:500), CD45 (catalog number ab25386, Abcam, 5 μg ml$^{-1}$). Secondary antibodies were antirabbit poly-horseradish peroxidase-IgG (included in kit) or rabbit antirat (catalog number A110-322A, Bethyl Laboratories, 1:250). Digital images of entire tissue sections were captured using the Leica Aperio AT2 digital scanner (×40, resolution 0.25 μM per pixel), viewed using the Leica Aperio Image Scope v.12.3.2.8013 and quantified by HALO (Indica Labs) image analysis (Supplementary Tables 28 and 33).

For immunofluorescence, tissue sections were incubated with primary antibodies: rhodamine-labeled DBA (catalog number RL-1032, Vector Laboratories, 1:100), rhodamine-labeled UEA I (catalog number RL-1062, Vector Laboratories, 1:100), ZSG (catalog number TA180002, Origene, 1:1,000), CK7 (catalog number ab181598, Abcam, 1:200), CK20 (catalog number ab97511, Abcam, 1:200), E-cadherin (catalog number AF748, R&D Systems, 1:100), N-cadherin (catalog number 13116, Cell Signaling Technology, 1:100), Icam1 (catalog number ab179707, Abcam, 1:100), Cdx2 (catalog number ab76541, Abcam, 1:100), Krt80 (catalog number 16835-1-AP, ProteinTech, 1:100), Hba-a1 (catalog number ab92492, Abcam, 1:100), Lgals3 (catalog number ab209344, Abcam, 1:200), CD45 (catalog number ab10558, Abcam, 1:200). Secondary antibodies included Alexa 488, 594 and 647 (catalog numbers A-11055, A-21207 and A-31571, Thermo Fisher Scientific, 1:500). Sections were counterstained (4,6-diamidino-2-phenylindole (DAPI); catalog number 4083, Cell Signaling, 1:10,000) and images captured using a Zeiss ImagerM2 and Apotome microscope or Zeiss Axioscan.Z1 (Zeiss) at ×40 magnification and processed using ZEN2.3 (Zeiss) software (Supplementary Tables 28 and 33). Single-channel images are shown in Supplementary Fig. 1.

Nalcn RNA expression was detected in formalin-fixed, paraffin-embedded sections using the Advanced Cell Diagnostics (ACD) RNAscope 2.5 LS Reagent Kit-RED (ACD, catalog number 322150) and RNAscope 2.5 LS Mm Nalcn (ACD, catalog number 415168). Probe hybridization and signal amplification were performed according to the manufacturer's instructions. Fast Red detection of mouse Nalcn was performed on the Bond Rx using the Bond Polymer Refine Red Detection Kit (Leica Biosystems, catalog number DS9390) according to the manufacturer's protocol. Whole-tissue sections were imaged on the Aperio AT2 (Leica Biosystems) and analyzed as for immunohistochemistry using HALO (Indica Labs) imaging analysis software. β-Galactosidase staining was performed exactly as described[4] (Supplementary Tables 26, 28 and 30).

Histological review, primary and metastatic tumor classification were performed by performed by expert pathologists (P. Vogel and B. Mahler-Araujo) blinded to mouse genotype and clinical history. The numbers of ZSG+ cell clusters or metastases were counted in each organ in each mouse. Tissue fibrosis was assessed by expert pathologist R. Nazarian using sections stained with H&E, Masson's trichrome and Picrosirius Red.

*Whole-tissue imaging.* Kidneys were exsanguinated, perfused with PBS and 4% PFA by PBS washes and immersion reagent 1a (150 g of ultrapure water, 20 g of Triton X-100 (catalog number 10254583, Thermo Fisher Scientific), 10 g of 100% solution of *N,N,N',N'*-tetrakis (2-hydroxypropyl)ethylenediamine (catalog number 122262, Sigma), 20 g of urea (catalog number 140750010, ACROS Organics), 1 ml of 5 M NaCl) containing 10 μM DAPI (catalog number 4083; Cell Signaling Technology) at 37 °C and 80 r.p.m. The solution was exchanged every 2 d until the tissue was cleared. Cleared tissues were washed and immersed in 50% PBS/50% reagent 2 (15 g of ultrapure water, 50 g of sucrose (catalog number 220900010, ACROS Organics), 25 g of urea (catalog number 140750010, ACROS Organics), 10 g of 2,2,2-nitrilotriethanol (catalog number 90279, Sigma)) for 6 h (room temperature, with gentle shaking) followed by immersion in 100% reagent 2 (10 ml) for 1 d (room temperature). Tissues were mounted and scanned on a TCS SP5 confocal laser scanning microscope (Leica) at ×10 objective for DAPI and endogenous expression of ZSG. Images were processed using Imaris x64 v.9.3.0 software (Oxford Instruments) (Supplementary Tables 26, 28 and 30).

Serial two-photon tomography imaging was performed on a TissueCyte 1000 instrument (TissueVision) in which a series of mosaic two-dimensional images are taken of the tissue, followed by physical sectioning with a vibratome and a subsequent round of imaging. This continues in an automated fashion, generating 15 μm serial two-photon tomography sections that can be mounted on standard microscopy slides, imaged by Axioscan fluorescence scanning (Zeiss) for section identification and realignment. Fiducial agarose marker beads labeled with GFP are distributed throughout the embedding medium to help in the realignment of the samples for consequent use (Supplementary Tables 26, 28 and 30).

**Harvesting and injection of circulating ZSG cells.** Peripheral blood (500 μl to 1 ml) was harvested from mice at autopsy into 10 μl of 0.5 M EDTA, diluted in PBS and assessed by MACSQuant Analyzer (Miltenyi Biotech Inc.) for ZSG expression (525/50 nm (FITC) versus 614/50 nm (propidium iodide)). Cells for SCS and tail-vein injection were sorted using a BD FACSAria II Cell Sorter (BD Biosciences) excitation at 525/50 nm (FITC) versus 614/50 nm (propidium iodide). Nontamoxifen-induced mouse peripheral blood served as a negative control to set gate parameters (Supplementary Figs. 2 and 3). Some 25,000 ZSG+ cells were sorted and injected into recipient NOD SCID gamma mice (Charles River) and aged. For serial dilution assessment of tCZC metastasis initiation, tCZCs were isolated from donor tumor-bearing animals via FACS based on ZSG expression and placed into culture medium. Culture medium was as follows: Advanced DMEM/F12 (catalog number 31330038, Thermo Fisher Scientific), 2 mM L-glutamine (catalog number 25030024, Thermo Fisher Scientific), B27 (catalog number 12587010, Thermo Fisher Scientific) and N2 (catalog number A1370701, Thermo Fisher Scientific), containing growth factors (50 ng ml⁻¹ epidermal growth factor (PeproTech), 100 ng ml⁻¹ basic fibroblast growth factor (catalog number 100-18c, PeproTech) and 1% FBS (catalog number 10500064, Thermo Fisher Scientific). Cells were grown at 37 °C in 5% CO₂. Recipient NOD SCID gamma mice (Charles River) were injected with either 10, 100, 1,000 or 10,000 tCZCs via tail-vein injection and aged. Full autopsy and tissue harvesting were performed as described above. Full autopsy and tissue harvesting were performed as described above (Supplementary Tables 26, 28 and 29).

**Bulk RNA sequencing.** Total RNA was extracted from tissues using Maxwell RSC miRNA Tissue Kit (catalog number AS1460, Promega). RNA quality was assessed using TapeStation System (catalog number 5067-5579, Agilent). RNA libraries and downstream sequencing were carried out as previously described[54]. The Illumina TruSeq stranded messenger RNA kit (catalog number 20020595, Illumina) was used to prepare RNA libraries and RNA quality confirmed using TapeStation (Agilent) and quantified using a KAPA qPCR library quantification kit for Illumina platforms (catalog number KK4873, KAPA Biosystems). Samples were normalized using the Agilent Bravo, pooled and sequenced on Illumina NovaSeq SP flowcell to generate single-end 50 bp reads at 20 million reads per sample.

Single-end 50 bp RNA reads were aligned to GRCm38 with HISAT2 (with default parameters). Each sample was sequenced across several lanes; per-lane BAM files were merged into per-sample BAM files. Quality control metrics were collected for each file, including duplication statistics and number of reads assigned to genes. Reads were counted on annotated features with subreads featureCounts, providing 'total', 'aligned to the genome' and 'assigned to a gene' (that is, included in the analysis) counts. Percentages of aligned bases were computed for several categories: coding, untranslated region, intronic and intergenic. Other quality control metrics were the percentage of reads on the correct strand, median coefficient of variation of coverage, median 5′ bias, median 3′ bias and the ratio of 5′ to 3′ coverage. Quality control also included an expression heatmap drawn using log₂-transformed counts. The log₂-transformed counts were generated from

normalized counts using the log2 function in R and counts function from DEseq2. Genes were regarded as displaying differential expression between sample cohorts if they displayed of ≥1 or ≤−1 log(fold difference) in expression levels with an adjusted *P* ≤ 0.05 (Supplementary Tables 26, 28 and 30).

**Single-cell RNA sequencing.** Animals were perfused with PBS followed by 100 U ml⁻¹ of collagenase type IV in HBSS with Ca²⁺ and Mg²⁺ (Life Technologies) media containing 3 mM CaCl₂. Whole organs were dissected, dissociated and placed into 2 ml of the appropriate dissociation buffer: lung and stomach were dissociated with 200 U ml⁻¹ of collagenase type IV (Sigma) and 100 μg μl⁻¹ of DNAse I (Roche) in HBSS with Ca²⁺ and Mg²⁺ (Life Technologies) media containing 3 mM CaCl₂; liver was dissociated with collagenase type I (100 U ml⁻¹), dispase (2.4 U ml⁻¹) DNAse I (100 μg ml⁻¹) in HBSS with Ca²⁺ and Mg²⁺ (Life Technologies) media containing 3 mM CaCl₂; kidney was dissociated with papain (20 U ml⁻¹) and DNAse I (100 mg ml⁻¹) in DMEM high glucose, 2 mM L-glutamine (Life Technologies) with 1× Pen-Strep and 10% FBS; uterus and epididymis were dissociated with collagenase type I (100 U ml⁻¹) and DNAse I (100 mg ml⁻¹) in in HBSS with Ca²⁺ and Mg²⁺ (Life Technologies) media containing 3 mM CaCl₂. Cells suspensions were filtered washed with HBSS without calcium and magnesium and centrifuged for 5 min at 300*g* at 4 °C for 5 min.

Single-cell suspensions of solid tissues were multiplexed and labeled with Cell Hashing conjugates: antimouse hashtags from 0301 to 0315 (BioLegend) before sequencing. All nucleated cells and ZSG⁺ cells isolated from peripheral blood were not multiplexed but placed into a 10x Genomics pipeline. SCS libraries were prepared using Chromium Single Cell 3′ Library & Gel Bead Kit v.3, Chromium Chip B Kit and Chromium Single Cell 3′ Reagent Kits v.3 User Guide (manual CG000183 Rev A; 10x Genomics). Cell suspensions were loaded on the Chromium instrument with the expectation of collecting gel-bead emulsions containing single cells. RNA from the barcoded cells for each sample was subsequently reverse-transcribed in a C1000 Touch thermal cycler (Bio-Rad) and all subsequent steps to generate single-cell libraries were performed according to the manufacturer's protocol with no modifications (for most of the samples 12 cycles was used for cDNA amplification, 16 for samples with very low cell concentration). cDNA quality and quantity were measured with Agilent TapeStation 4200 (High Sensitivity D5000 ScreenTape) after which 25% of material was used for preparation of the gene expression library. Library quality was confirmed with Agilent TapeStation 4200 (High Sensitivity D1000 ScreenTape to evaluate library sizes) and Qubit 4.0 Fluorometer (Qubit dsDNA HS Assay Kit (Thermo Fisher Scientific) to evaluate double-stranded DNA quantity). Each sample was normalized and pooled in equal molar concentrations. To confirm concentration pools underwent qPCR using KAPA Library Quantification Kit on QuantStudio 6 Flex before sequencing. Pools were sequenced on an Illumina NovaSeq6000 sequencer with the following parameters: 28 bp, read 1; 8 bp, i7 index; and 91 bp, read 2.

Raw RNA reads were processed with cellranger using mm10 from 10x as the reference genome to create filtered gene expression matrixes. Cell barcodes detected by cellranger were used as input to CITESeq for hashtagged sequence data (solid organs) generating a counts matrix with cell barcodes and hashtag oligo sequences per cell. The HTODemux function from Seurat was then used to identify clusters and classify cells according to their barcodes, including negative and doublet cells. Quality control metrics were generated using Scater followed by single-cell object conversion to Seurat objects, merging of objects and then analyses run using the standard Seurat pipeline (Supplementary Tables 26, 28 and 30).

SCS profiles of human CTCs (GSE75367; GSE74639; GSE60407; GSE67980; GSE114704; GSE144494) and 500 cells from Illumina 10x for human PBMC raw counts were merged in python v.3.7.3 using the pandas library. Only common genes between datasets were analyzed. Seurat objects were created from PBMCs and CTCs. Following this step, data were analyzed using the standard Seurat pipeline (Supplementary Table 33).

For direct comparison of human CTCs and mouse tCZCs, 15,328 orthologs were identified and profiles processed through the standard Seurat workflow that includes a per-cell normalization of each gene expression count. Enrichment of a hemoglobin gene expression was carried out in UCell and enrichment scores generated with a two-tailed Mann–Whitney *U* statistic.

**Statistics and reproducibility.** Clinical and mutation/CNA data were from the Cancer Genome Atlas via cBioportal, selecting for studies included as part of the Pan-Cancer Atlas[55]. Gene expression data was from Xenabrowser[56]. d*N*/d*S* was calculated using the dNdScv R package[18]. *t*-distributed stochastic neighbor embedding (*t*-SNE) was performed with the sklearn library in python using the 'BarnesHut' method, with a perplexity of 15, learning rate of 1,000 and 1,000 iterations. Contours were drawn as kernel density estimates of the density of nonsynonymous *NALCN* mutations for each cancer subtype with a significant (*P* ≤ 0.05) d*N*/d*S* score individually.

NALCN cryo-electron microscopy[12] structure 6XIW was downloaded from pdb, simulated in MemprotMD[22,23], energy minimized using the steepest descents for 5,000 steps and converted to a MARTINI coarse-grained[57] representation embedded in a 1-palmitoyl-2-oleoyl-*sn*-glycero-3-phosphocholine bilayer with 575 lipids. The membrane was self-assembled by position restraining the protein

and simulating for 200 ns to allow the membrane to form. The CG system was simulated for 1,000 ns before converting back to atomistic detail using CG2AT. The resultant atomistic membrane system was simulated in fully atomistic detail using the gromos53a6 forcefield for 400 ns. Mutational impact on pore size was calculated using HOLE[21]. Mutations were introduced into the simulated NALCN structure using the modeler mutation optimization protocol, and the resultant HOLE pore profiles aligned on their selectivity filters (Supplementary Table 28).

Spatial clustering of mutations was performed by calculating the distance between the center of mass of each pair of mutated residues (in the wild-type structure), and grouping residues into clusters with distances between any one pair of residues <12 Å. We calculated an expected distribution through randomly sampling the structure for the same number of mutations observed overall 100,000 times. Comparison of the observed clusters with the distribution of random samples was used to calculate a *P* value. All code generated for spatial clustering and analysis, with a workable example is available at: https://github.com/shorthouse-mrc/NALCN.

Unsupervised hierarchical clustering was performed using Morpheus (https://software.broadinstitute.org/morpheus) and genset enrichment using g:Profiler (version e104_eg51_p15_3922dba). Tissue type deconvolution was performed using xCell (http://xCell.ucsf.edu/) (Supplementary Table 28).

**Reporting summary.** Further information on research design is available in the Nature Research Reporting Summary linked to this article.

## Data availability
All the raw sequencing data have been deposited in the Gene Expression Omnibus with the following accession numbers: mouse RNA-seq of tumors and metastases (GSE210134) and mouse single cell RNA-seq of CZCs, PBMCs, tumors, metastases and solid tissues (GSE210134). Murine Prom1+ gastric mucosa and adenocarcinoma data GEO accession number: GSE78076. NALCN mutation and TSNE plot were generated with Pan-Cancer Atlas data from TCGA via cbioPortal and Xenabrowser. Cancer staging data were generated with Pan-Caner Atlas from TCGA and COSMIC data. NALCN structure 6XIW was from pdb. Human CTC and gene signature datasets are from the following GEO accession numbers: GSE75367, GSE74639, GSE60407, GSE67980, GSE114704, GSE144494. Human PBMC data are from Illumina 10x (10k Human PBMCs, 3' v3.1, Chromium X). Source data are provided with this paper.

## Code availability
All bulk and single-cell analyses and visualizations were performed using R software (v.3.6.1), R studio (v.1.3.1093) and Python (v.3.9). Details on specific packages are included in the Methods section and also at https://github.com/shorthouse-mrc/NALCN. No custom code was used for any part of the data processing or analysis.

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

## Acknowledgements
This work was supported by grants to: R.J.G. (CRUK Major Centre and Cambridge Institute Core Awards and St. Jude Children's Research Hospital/American Lebanese Syrian Associated Charities (ALSAC)); S.S.Z. (ALSAC, National Institutes of Health CA021765, MH097742, DC012833); B.A.H. (Royal Society Research Fellowship UF130039, Medical Research Council research grant MR/S000216/1); E.P.R. (Marie Skłodowska-Curie Individual Fellowship from European Commission and Cancer Center Neurobiology and Brain Tumor Program Garwood Named Fellowship from St. Jude Children's Research Hospital). We thank the CRUK Cambridge Institute genomics, flow cytometry, light microscopy, biological resource unit, bioinformatics, research instrumentation and cell services, histopathology core facilities and the cell and tissue imaging resources and the Animal Resources Center at St. Jude Children's Research Hospital for technical assistance.

## Author contributions
E.P.R. conducted the great majority of the experiments, contributed to the conception of the research, experimental design and writing of the manuscript. D.S., A.J., L.P.H., M.O., M.P.-R., A.F., G.J.H., R.T., F.C.L., J.A.B., B.A.H., S.S.Z., D.J.W., L.Z. and J.K. conducted important experiments, provided critical advice on experimental design and/or interpretation and contributed to writing of the manuscript. B.M.-A., P.V. and R.M.N. provided pathology expert review and contributed to writing of the manuscript. R.J.G. conceived the research and experimental design, supervised the research and was primary author of the manuscript.

## Competing interests
The authors declare no competing interests.

## Additional information
**Extended data** is available for this paper at https://doi.org/10.1038/s41588-022-01182-0.

**Correspondence and requests for materials** should be addressed to Richard J. Gilbertson.

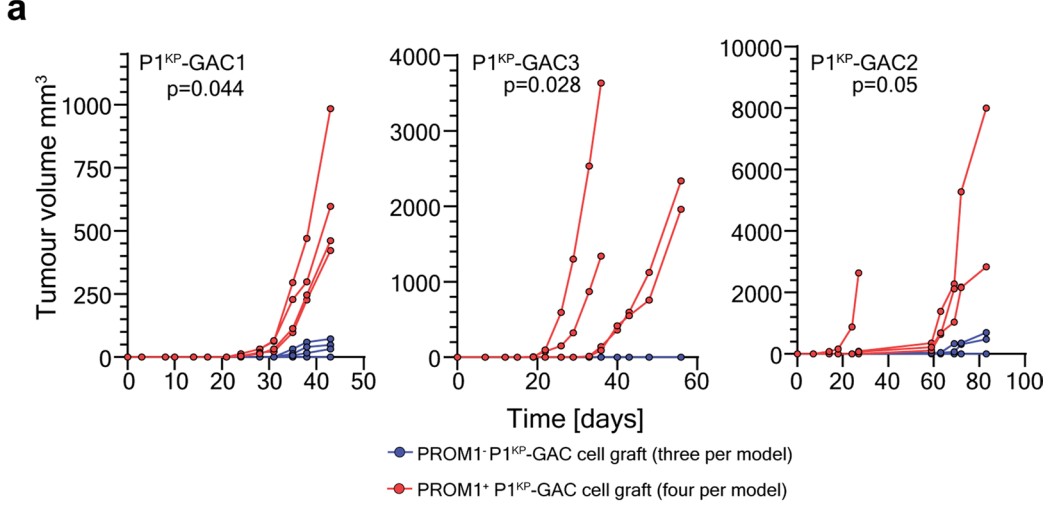

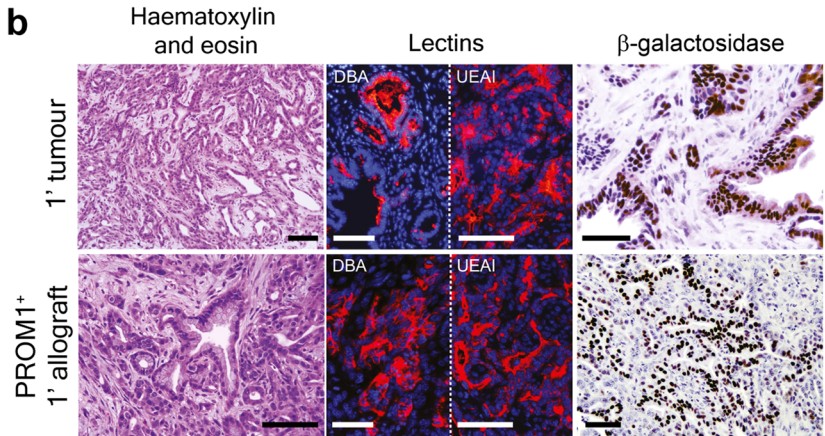

**Extended Data Fig. 1 | Prom1+ *P1^KP*-GAC cells propagate gastric adenocarcinoma as allografts. (a)** Growth of allografts derived from three independent *P1^KP* gastric adenocarcinomas (GAC) in immunocompromised mice (PROM1+ n=4 and PROM1−n=3 for each allograft). Statistics are based on Permutation test whereby permuted p-values compare PROM1+ and PROM1− allografts for each tumour type using the average t-statistic between the groups as the test statistic. Two-tailed Permutation test GAC1 p=0.044, GAC2 p=0.05, GAC3 p=0.028. **(b)** Histology of primary *P1^KP*-GAC and daughter PROM1+ cell allografts. Tumours expressed gastric adenocarcinoma-associated lectins and PROM1 identified by Beta-galactosidase staining from *Prom1 ^CreERT2/LacZ* locus (scale=100um). Representative photomicrographs Primary *P1^KP*-GAC n=3 and PROM1+ cell allografts n=12 evaluated histologically.

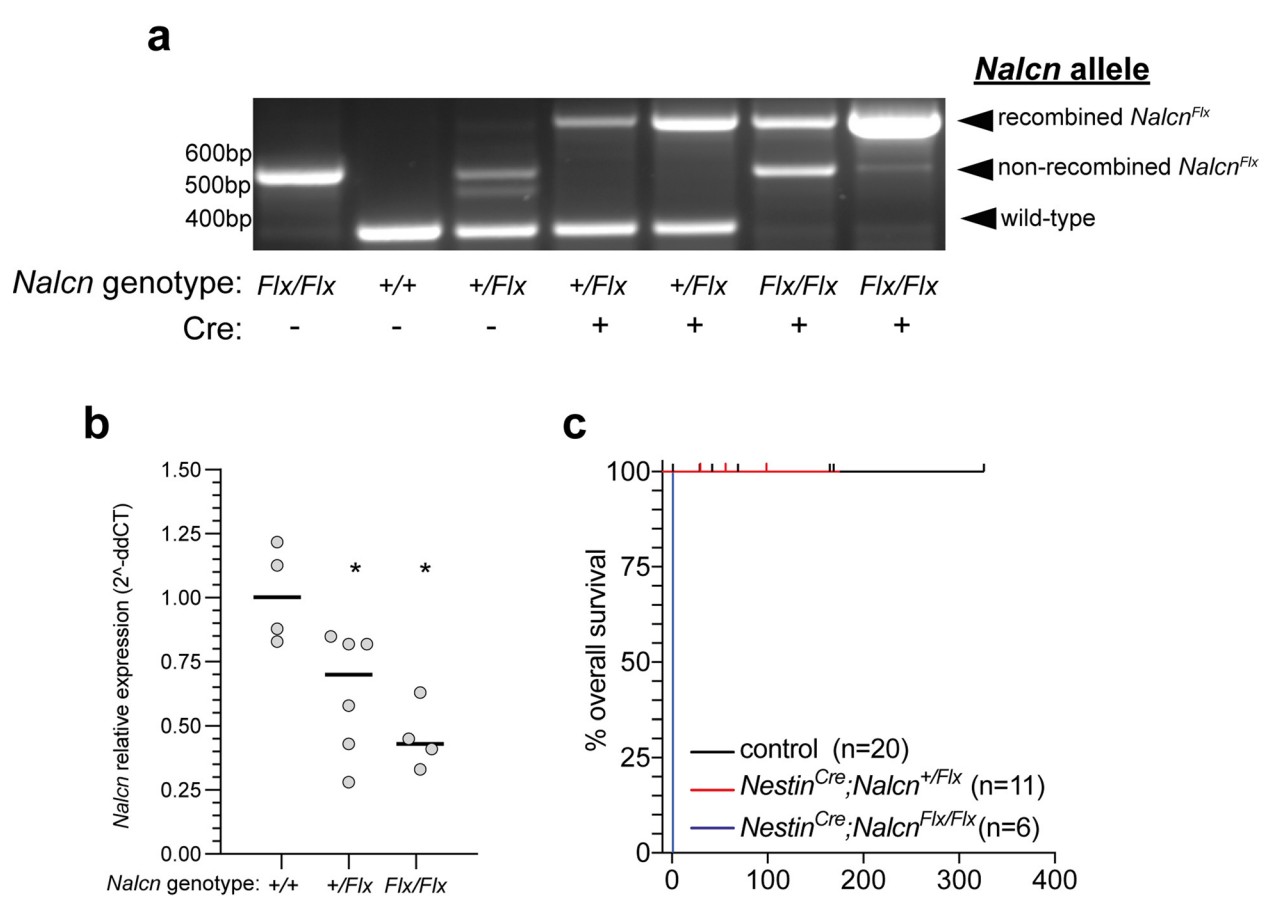

**Extended Data Fig. 2 | Recombination of the *Nalcn^Flx* conditional allele deletes the gene *in vivo*. (a)** Polymerase chain reaction (PCR) products derived from brains of *Nalcn^+/+*, *Nalcn^+/Flx*, or *Nalcn^Flx/Flx* mice with or without the *Nestin^Cre* allele. **(b)** Real-time reverse transcriptase PCR analysis of *Nalcn* RNA expression in brains of *Nalcn^+/+* (n = 4), *Nalcn^+/Flx* (n = 6), or *Nalcn^Flx/Flx* (n = 4) mice carrying the *Nestin^Cre* allele. bar = median (*p = 0.0190 *Nalcn^+/Flx*, *p = 0.0286 *Nalcn^Flx/Flx*; two-tailed Mann-Whitney U Test). **(c)** Similar to germline deletion of *Nalcn*, homozygous deletion of *Nalcn* from the brains of *Nestin^Cre*;*Nalcn^Flx/Flx* (n = 6) mice was lethal at birth due to respiratory arrest, providing functional validation of *Nalcn* deletion. control (n = 20) and *Nestin^Cre*;*Nalcn^+/Flx* (n = 11).

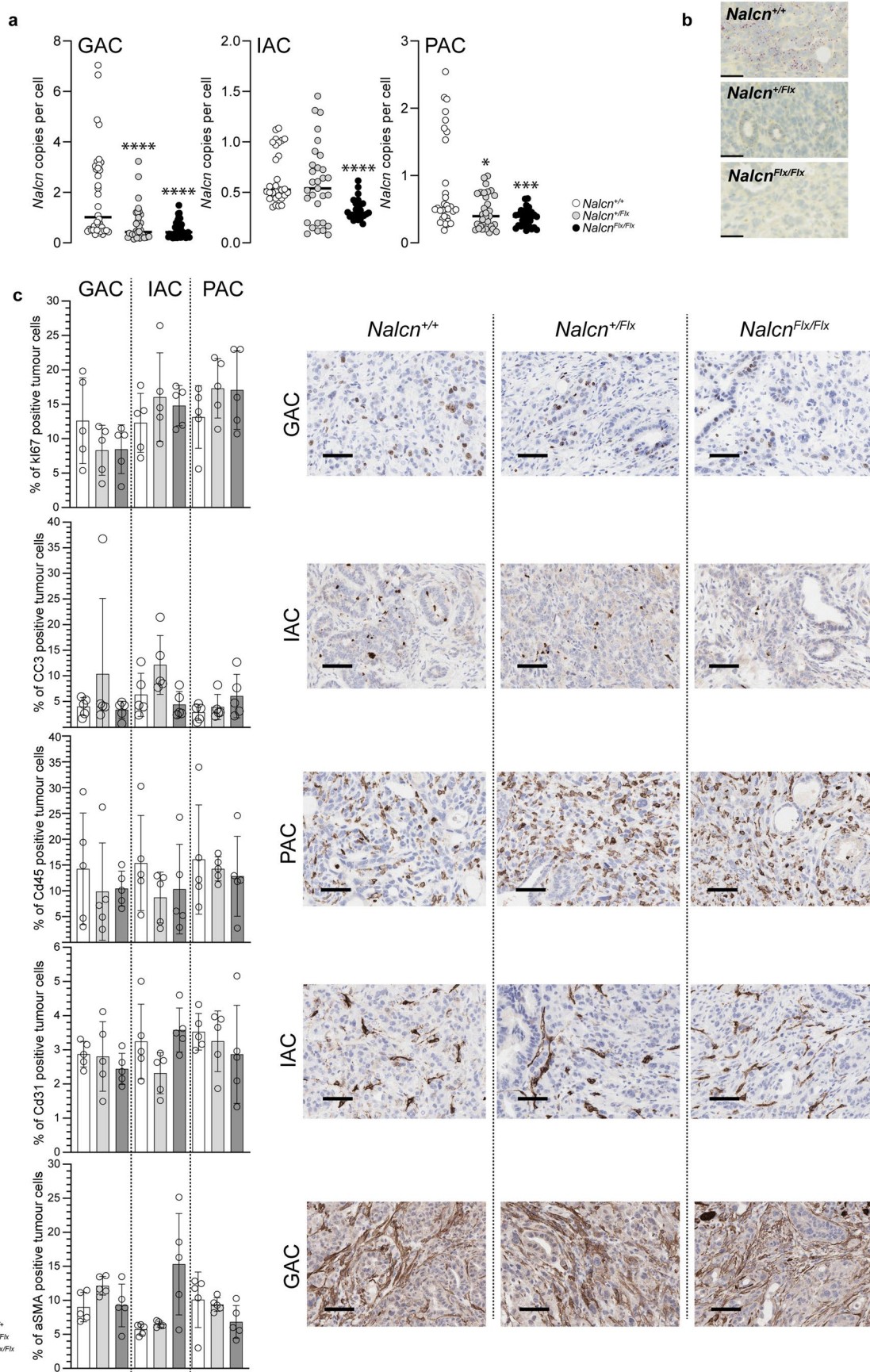

**Extended Data Fig. 3 | See next page for caption.**

**Extended Data Fig. 3 |** *Nalcn* **deletion does not affect cell proliferation, apoptosis, immune-infiltration, vasculature or ASMA expression in primary tumours in** *P1^{KP}*, *V1^{KP}* **or** *Pdx1^{KP}* **mice. (a)** HALO-quantification of *Nalcn* mRNA transcripts per cell detected by RNA-scope analysis in mouse gastric (GAC), intestinal (IAC) and pancreatic (PAC) adenocarcinomas of the indicated *Nalcn* genotype (bar=median; *p = 0.0294; ***p = 0.0004; ****=p < 0.0001, two-tailed Mann-Whitney test). Data are tumour fields (5–8 images per tumour) from n = 5 tumours for each *Nalcn* genotype of *P1^{KP}*-GAC, *V1^{KP}*-IAC and *Pdx1^{KP}*-PAC mice (total n = 45 unique tumours, 289 unique tumour fields). **(b)** Representative photomicrographs of *Nalcn* RNA *in situ* hybridization in GACs (n = 15 biologically distinct tumours, 100 tumour fields) of the indicated *Nalcn* genotype (scale=50 μm). **(c)** Left in each is HALO-quantification (Data are mean ± SD) of immunohistochemically-detected tumour cell expression of MKI67 (proliferation), cleaved Caspase-3 (CC3; apoptosis), CD45 (immune cell infiltration), CD31 (endothelial cells) and alpha-smooth muscle actin ASMA; stroma) in five complete biologically independent tumour fields for each *Nalcn* genotype of *P1^{KP}*-GAC, *V1^{KP}*-IAC and *Pdx1^{KP}*-PAC mice (total n = 45 unique tumours). Two-tailed Mann-Whitney U tests revealed no significant difference in these markers among tumours with different *Nalcn* genotypes. P-values GAC, IAC, PAC of *Nalcn^{+/+}* vs *Nalcn^{+/Flx}*, *Nalcn^{+/+}* vs *Nalcn^{Flx/Flx}*, respectively: KI67 (0.4206, 0.4206, 0.4206, 0.5476, 0.2222, 0.5476), CC3 (0.9999, 0.5476, 0.0952, 0.5476, 0.9999, 0.2222), CD45(0.6905, 0.8413, 0.1508, 0.3095, 0.6905, 0.8413), ASMA(0.0556, 0.8413, 0.3095, 0.0556, 0.2222, 0.1508), CD31(0.9999, 0.0952, 0.0952, 0.4206, 0.8413, 0.1508). Right in each are exemplar photomicrographs of the indicated marker in the indicated tumour type (scale=50um).

**a**

### $P1^{KP}$-GAC metastases

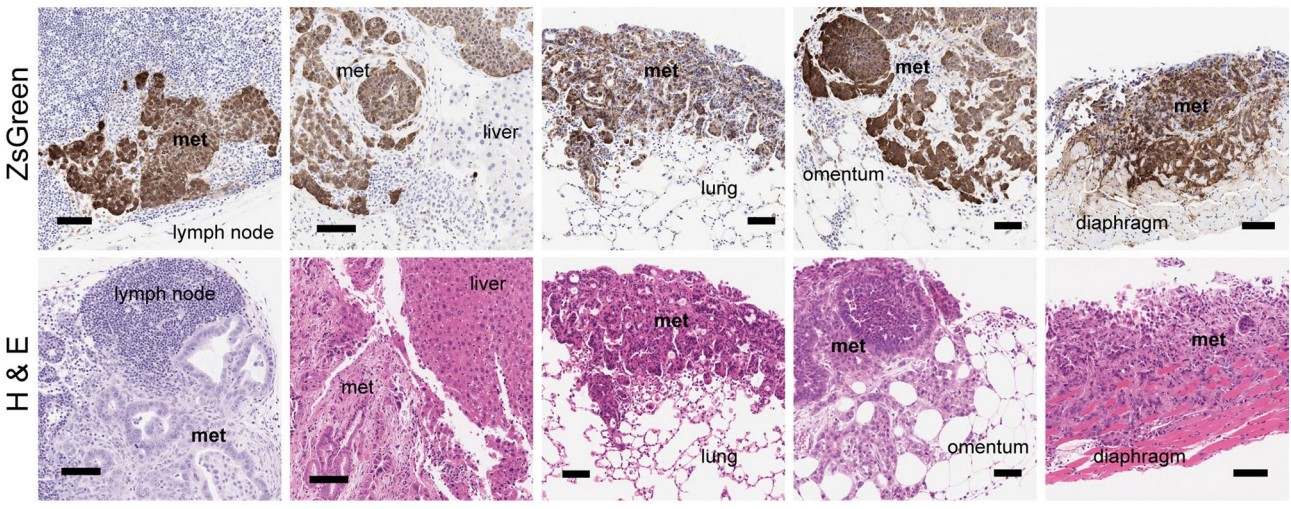

**b**

### $V1^{KP}$-IAC metastases

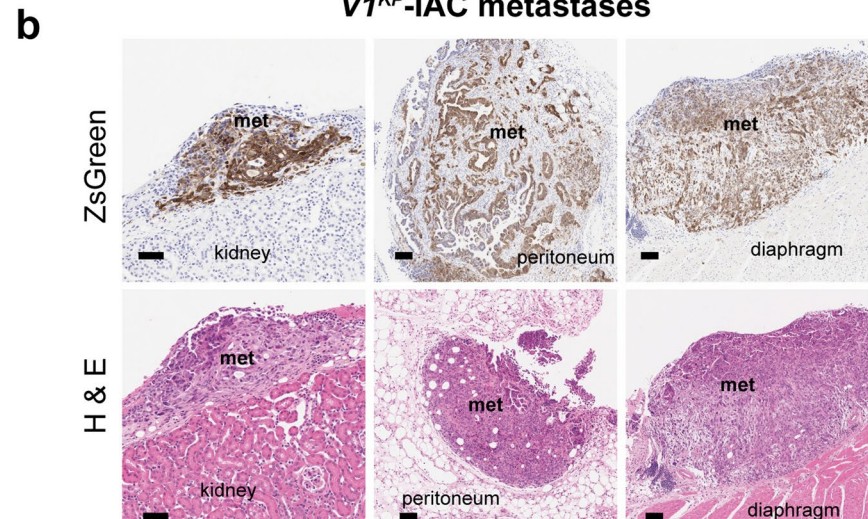

**c**

### $Pdx1^{KP}$-PAC metastases

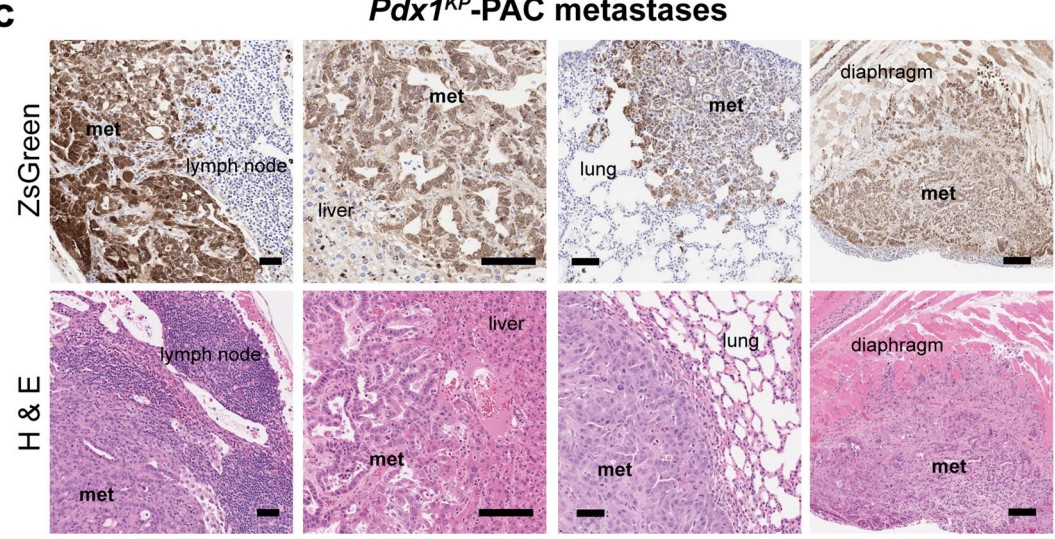

**Extended Data Fig. 4 | Metastases of $P1^{KP}$-GAC, $V1^{KP}$-IAC and $Pdx1^{KP}$-PAC.** Photomicrographs of **(a)** $P1^{KP}$-GAC, **(b)** $V1^{KP}$-IAC and **(c)** $Pdx1^{KP}$-PAC metastases to the indicated tissues. Top in each, immunohistochemistry of ZsGreen staining. Bottom in each, haematoxlin and eosin (H & E) stain (scale=100um). All enumerated metastases were evaluated by H&E (full list Supplementary Table 7; n = 7,076 metastases) n = 59 metastases evaluated by ZSG for IHC.

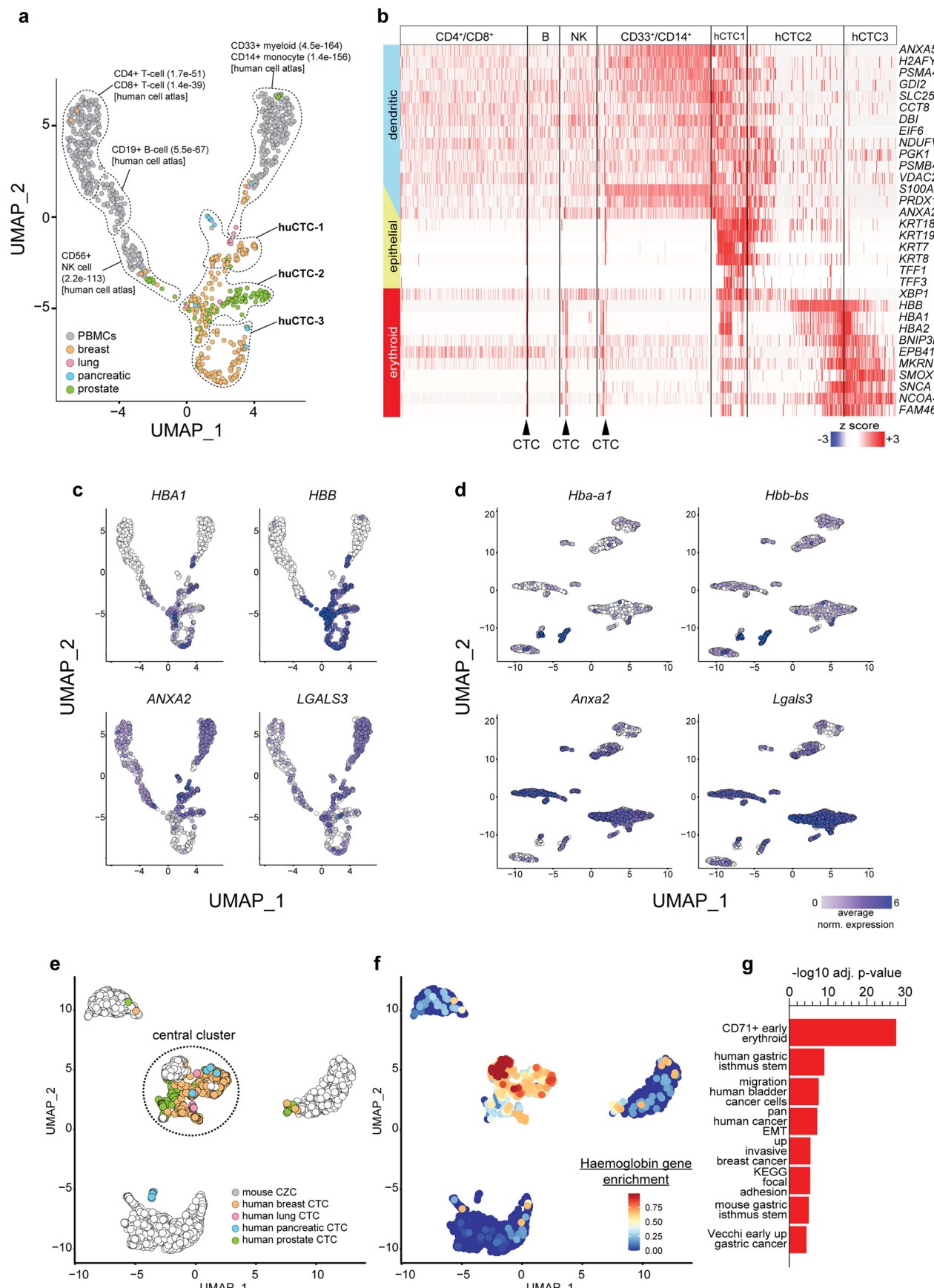

**Extended Data Fig. 5 | See next page for caption.**

**Extended Data Fig. 5 | Human circulating tumour cells (CTCs) and peripheral blood mononuclear cells (PBMCs). (a)** UMAP of single cell RNA sequencing (SCS) profiles of human CTCs and PBMCs (see main text for SCS sources). Genesets enriched in the indicated SCS clusters are shown with adjusted p-value for enrichment in parenthesis. **(b)** Heatmap of indicated gene expression from relevant genesets enriched in each cell from each cluster in (a). **(c)** Feature plots of exemplar genes enriched in human CTCs in (a). **(d)** Mouse orthologues of human genes in (c) mapped onto the UMAP of mouse CZCs and PBMCs in main Fig. 3b. **(e)** UMAPs of SCS profiles of common orthologues expressed in human CTCs and mouse tCZCs. **(f)** Enrichment of haemoglobin gene expression in UMAP shown in (e). **(g)** Geneset enrichments in the dotted-line enclosed, central cluster relative to the other SCS profiles is reported in **(e)**.

**a**

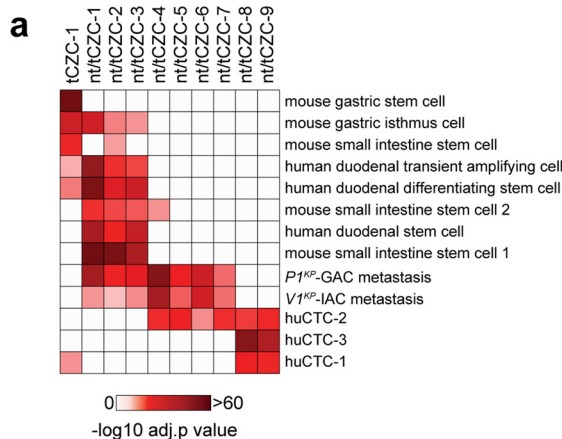

|  | tCZC-1 | nt/tCZC-1 | nt/tCZC-2 | nt/tCZC-3 | nt/tCZC-4 | nt/tCZC-5 | nt/tCZC-6 | nt/tCZC-7 | nt/tCZC-8 | nt/tCZC-9 |  |
|---|---|---|---|---|---|---|---|---|---|---|---|
| mouse gastric stem cell |  |  |  |  |  |  |  |  |  |  | |
| mouse gastric isthmus cell |  |  |  |  |  |  |  |  |  |  | |
| mouse small intestine stem cell |  |  |  |  |  |  |  |  |  |  | |
| human duodenal transient amplifying cell |  |  |  |  |  |  |  |  |  |  | |
| human duodenal differentiating stem cell |  |  |  |  |  |  |  |  |  |  | |
| mouse small intestine stem cell 2 |  |  |  |  |  |  |  |  |  |  | |
| human duodenal stem cell |  |  |  |  |  |  |  |  |  |  | |
| mouse small intestine stem cell 1 |  |  |  |  |  |  |  |  |  |  | |
| $P1^{KP}$-GAC metastasis |  |  |  |  |  |  |  |  |  |  | |
| $V1^{KP}$-IAC metastasis |  |  |  |  |  |  |  |  |  |  | |
| huCTC-2 |  |  |  |  |  |  |  |  |  |  | |
| huCTC-3 |  |  |  |  |  |  |  |  |  |  | |
| huCTC-1 |  |  |  |  |  |  |  |  |  |  | |

0 ▭ >60

-log10 adj.p value

**b**                                                    **c**

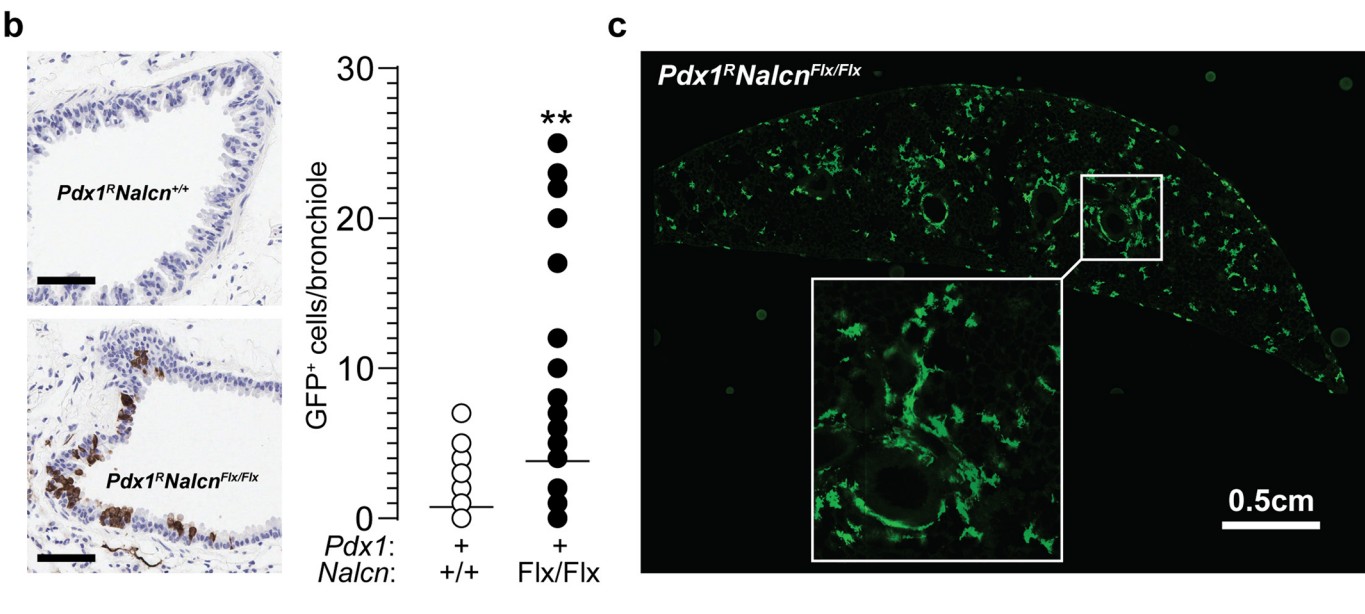

**d**

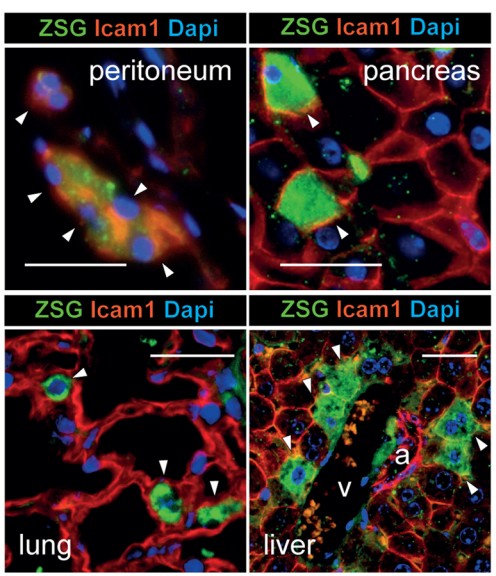

**Extended Data Fig. 6 | See next page for caption.**

**Extended Data Fig. 6 | NALCN loss-of-function circulating non-tumour cells (ntCZCs) resemble human and mouse CTCs and embed in distant organs.**
**(a)** Heatmap reporting geneset enrichment analysis in the UMAP clusters identified in main Fig. 7b. Test Genesets were derived from 2,086 different tissue and cell types including bulk RNAseq of mouse normal tissues and tumours, huCTC signatures, and mouse and human intestinal stem and mature cell signatures (see Methods). **(b)** ZSG immunohistochemistry of aged *Pdx1$^R$Nalcn$^{+/+}$* (top left) and *Pdx1$^R$Nalcn$^{Flx/Flx}$* (bottom left) mouse lung bronchioles (scale=100um). Right, the number of ZSG$^+$ cells/bronchiole in the lungs of *Pdx1$^R$Nalcn$^{+/+}$* (n = 2 mice, 6 lung lobes, 121 bronchiole) and *Pdx1$^R$Nalcn$^{Flx/Flx}$* (n = 1 mouse, 4 lung lobes, 57 bronchioles). (bar=median; **p = 0.0051 two-tailed Mann-Whitney U Test). **(c)** Two-photon direct ZSG$^+$ cell clusters detected in entire lung section of a *Pdx1$^R$Nalcn$^{Flx/Flx}$* mouse. **(d)** Exemplar co-immunofluorescence of tail vein injected *P1$^R$Nalcn$^{Flx/Flx}$* ntCZCs (arrows) incorporated into the organs of recipient mice (arrows indicated ZSG$^+$ cells, scale bar=50um).

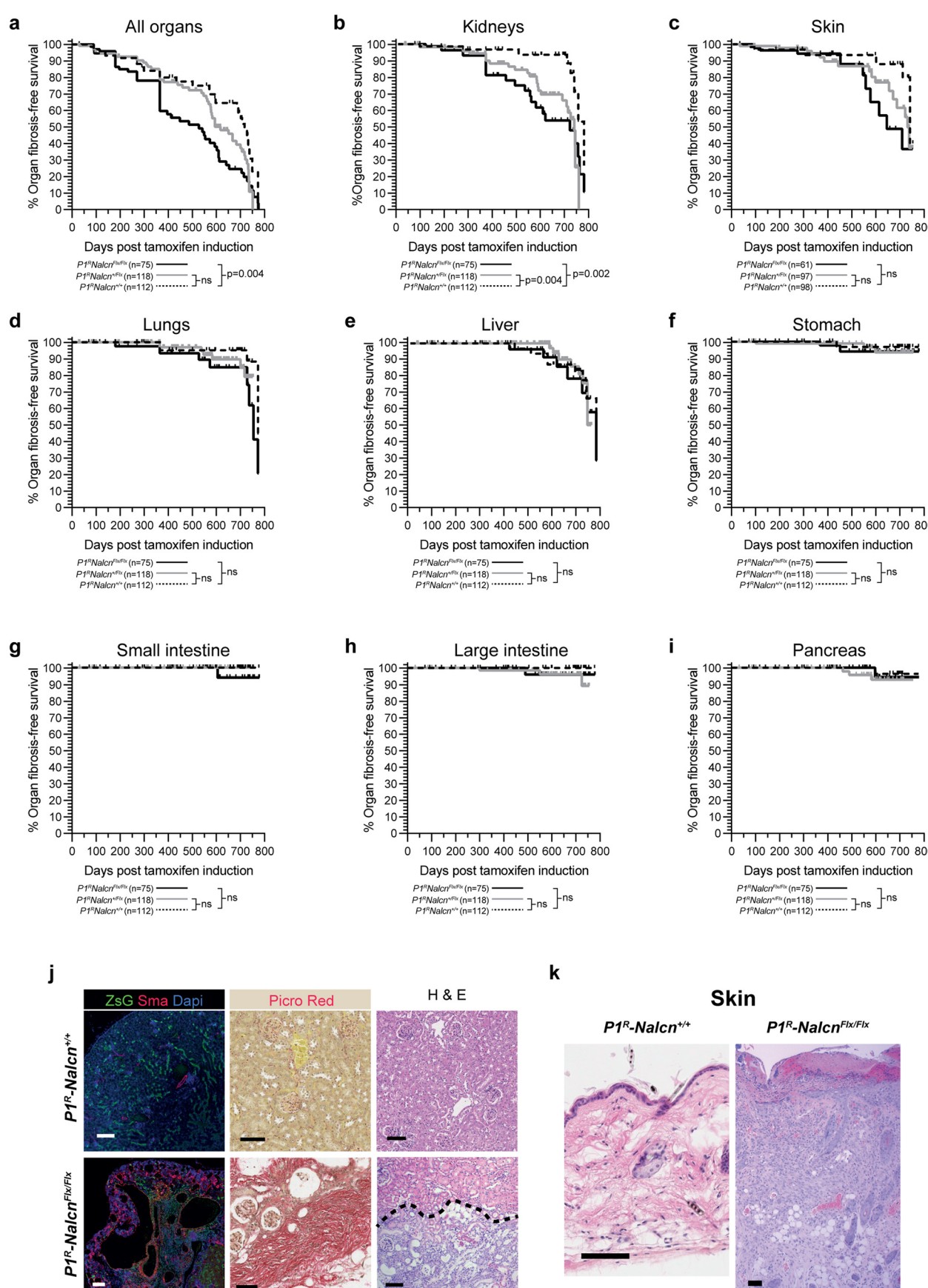

**Extended Data Fig. 7 | See next page for caption.**

**Extended Data Fig. 7 | Organ fibrosis following conditional deletion of *Nalcn* at P3 in *P1$^R$* mice.** Fibrosis-free survival for all organs **(a)** or the indicated organs **(b-i)**. P value reports the log-rank statistic (Mantel-Cox). The numbers of animals of each genotype are shown. p-values for each graph comparing *P1$^R$Nalcn$^{+/+}$* and *P1$^R$Nalcn$^{+/Flx}$* and *P1$^R$Nalcn$^{+/+}$* and *P1$^R$Nalcn$^{Flx/Flx}$*, respectively: All organs (0.0664, 0.0035), Kidneys (0.0037, 0.0022), Skin (0.1195, 0.0569), Lungs(0.3791, 0.1000), Liver(0.8846, 0.7250), Stomach(0.4938, 0.4225), Small intestine(>0.9999, 0.2348), Large intestine(0.1312, 0.2655), Pancreas(0.4764, 0.7571). **(j)** Photomicrographs of haematoxlin and eosin (H & E) and Picro-Sirus Red stain and co-immunofluorescence of ZsGreen, alpha-smooth muscle actin (ASMA) and Dapi in kidney from *P1$^R$Nalcn$^{+/+}$* and *P1$^R$Nalcn$^{Flx/Flx}$* mice aged >400 days. Note in *P1$^R$Nalcn$^{Flx/Flx}$* kidney: extensive fibrosis below the hashed line (H & E); marked Picro-Sirius Red staining indicating extensive fibrosis; ZsGreen recombination, gross distortion of normal kidney architecture and extensive alpha-SMA expression. **(k)** Photomicrographs of H & E stained skin from *P1$^R$Nalcn$^{+/+}$* and *P1$^R$Nalcn$^{Flx/Flx}$* mice aged >400 days. Note in *P1$^R$Nalcn$^{Flx/Flx}$* skin: ulceration and thickening of cornified layer, marked thickening of squamous cell layer and fibrosis of dermal layer. (scale 100um).

# Reporting Summary

## Statistics

For all statistical analyses, confirm that the following items are present in the figure legend, table legend, main text, or Methods section.

| n/a | Confirmed | |
|---|---|---|
| ☐ | ☒ | The exact sample size (*n*) for each experimental group/condition, given as a discrete number and unit of measurement |
| ☐ | ☒ | A statement on whether measurements were taken from distinct samples or whether the same sample was measured repeatedly |
| ☐ | ☒ | The statistical test(s) used AND whether they are one- or two-sided<br>*Only common tests should be described solely by name; describe more complex techniques in the Methods section.* |
| ☐ | ☒ | A description of all covariates tested |
| ☒ | ☐ | A description of any assumptions or corrections, such as tests of normality and adjustment for multiple comparisons |
| ☐ | ☒ | A full description of the statistical parameters including central tendency (e.g. means) or other basic estimates (e.g. regression coefficient) AND variation (e.g. standard deviation) or associated estimates of uncertainty (e.g. confidence intervals) |
| ☐ | ☒ | For null hypothesis testing, the test statistic (e.g. *F*, *t*, *r*) with confidence intervals, effect sizes, degrees of freedom and *P* value noted<br>*Give P values as exact values whenever suitable.* |
| ☒ | ☐ | For Bayesian analysis, information on the choice of priors and Markov chain Monte Carlo settings |
| ☐ | ☒ | For hierarchical and complex designs, identification of the appropriate level for tests and full reporting of outcomes |
| ☒ | ☐ | Estimates of effect sizes (e.g. Cohen's *d*, Pearson's *r*), indicating how they were calculated |

*Our web collection on statistics for biologists contains articles on many of the points above.*

## Software and code

Policy information about availability of computer code

| | |
|---|---|
| Data collection | MACSQuant software (version 10), Microsoft Excel (version 16.49), TCGA via cBioportal, Xenabrowser, HALO v2.0 |
| Data analysis | No custom code was used for any part of the data processing or analysis. Log-Rank (Mantel-Cox) test, Mann Whitney U two-tailed tests and multi-variate analyses were performed using Prism (version 10). RNAsequencing data (human and mouse, bulk and single cell) and dN/dS were done using R software (3.6.1), R studio (1.3.1093),and python (3.9) (see methods for details). g:GOSt for gene enrichment analysis by cumulative hypergeometric probability and multiple testing correction. MORPHEUS for pair-wise comparison analyses and hierarchical clustering, https://software.broadinstitute.org/morpheus. Code used for analyses can be found here: https://github.com/shorthouse-mrc/NALCN |

For manuscripts utilizing custom algorithms or software that are central to the research but not yet described in published literature, software must be made available to editors and reviewers. We strongly encourage code deposition in a community repository (e.g. GitHub). See the Nature Portfolio guidelines for submitting code & software for further information.

## Data

Policy information about availability of data

All manuscripts must include a data availability statement. This statement should provide the following information, where applicable:
- Accession codes, unique identifiers, or web links for publicly available datasets
- A description of any restrictions on data availability
- For clinical datasets or third party data, please ensure that the statement adheres to our policy

All the raw sequencing data have been deposited in the Gene Expression Omnibus with the following accession numbers: mouse RNA-seq of tumours and

# Field-specific reporting

Please select the one below that is the best fit for your research. If you are not sure, read the appropriate sections before making your selection.

☒ Life sciences ☐ Behavioural & social sciences ☐ Ecological, evolutionary & environmental sciences

For a reference copy of the document with all sections, see nature.com/documents/nr-reporting-summary-flat.pdf

# Life sciences study design

All studies must disclose on these points even when the disclosure is negative.

| | |
|---|---|
| Sample size | Sample sizes for each experiment are indicated in the figures, figure legends or main text. No statistical methods were used to predetermine sample size. The sample sizes for animal experiments were chosen based on previous experience in the lab and from published works specifically on tumour penetrance and latency. Zhu et al 2016, Jackstadt et al 2019, Hingorani et al 2003. Samples from single cell RNAseq generated thousands of cells which were acquired across several different genetic cohorts. Aside from animal experiments, we executed a minimum of three biological replicates whether it be for in vitro cell culture based assays, histological analyses, or transciptomic analyses. In some cases where only three biological replicates were done, it was dictated by availability of materials. |
| Data exclusions | Cell selection and filtering criteria for the single cell RNAseq analyses are detailed in the Methods section. Quality control metrics were generated using Scater. We removed cells on a per sample basis which had greater than 3 MADs (median absolute deviations) above the mean percentage of mitochondrial content, cells which had greater or fewer genes detected than 3 MADs above or below the mean and cells which had greater or fewer UMIs than 3 MADs above or below the mean. Genes which were detected in less than 5 cells (across all samples) were then removed from the matrix. Single cell objects filtered according to criteria above were generated from this step. Single cell objects were then converted to Seurat objects using the as.Seurat function. Doublet and negative cells were removed from objects representing solid organs (hash-tagged data). Multiple Seurat objects representing different samples were merged together using the merge function before a universal filtering step was carried out. Cells were filtered based on a threshold of cells with more than 100 and less than 5000 genes and less than 20% mitochondrial content. |
| Replication | All experiments were performed in at least 3 biologically independent replicates. All replicates on which statistics were carried out are biological replicates. In the case of genetically engineered mouse models, studies were balanced for age and sex. For the single cell RNAseq, cell types were reliably identified across multiple biological replicates. All attempts at replication of the results were successful. |
| Randomization | For in vivo work, induction of genetic alterations occurred either spontaneously embryonically or induced with tamoxifen exposure at postnatal day 3 or 60. Animals were chosen based on correct genotypes based on presence of a cre-recombinase allele (Nestin-cre, Pdx1-cre, villin1-CreERT2, Prom1-CreERT2, Rosa26-CreERT2), a reporter allele (Rosa26-ZSGreen), one of 3 combinations of Nalcn allele (wt, het, null) and then optionally combinations of oncogenic alleles for tumour studies (KrasG12D, Trp53flx). Sex-specific differences were minimized by including similar number of male and female animals. Each experiment contained animals from several different litters over the course of 2 years. For aging tumour studies, randomization was not applicable. For the gadolinium induciton experiments, all littermates were injected with gadolinium to ensure we captured representation of all Nalcn allele combinations. <br><br>In allograft experiments we utilized female mice from 4-16 weeks of age (NSG, Foxn1) that were purchased from Charles River. Age range was due to availability of primary material (circulating ZSG+ cells) from donor animals. Again, randomization was not applicable for these studies. <br><br>For studies where gastric spheres were expose to 4-hydroxytamoxifen, 8 wells of a 12-well plate were seeded with syngenic gastric organoids. 4 wells were chosen on the plate to receive 4-hydroxytamoxifen which in which 2 wells from each row were choosen to balance plating positioning effects. |
| Blinding | Multiple investigators were involved with sample collection, annotation, observational recordings and analysis and all were blinded to animal genotypes. For gastric sphere assays counting, electrophysiology researchers were blinded to the conditions for data collection and image analysis. For blood sample analyses, at time of blood draw only the animal ID was known and nothing about the genotype. Samples were analyzed and data collected prior to de-identifying. For metastasis enumeration, all data were collected without knowledge of the animal genotype. |

# Reporting for specific materials, systems and methods

We require information from authors about some types of materials, experimental systems and methods used in many studies. Here, indicate whether each material, system or method listed is relevant to your study. If you are not sure if a list item applies to your research, read the appropriate section before selecting a response.

## Materials & experimental systems

| n/a | Involved in the study |
|---|---|
| ☐ | ☒ Antibodies |
| ☐ | ☒ Eukaryotic cell lines |
| ☒ | ☐ Palaeontology and archaeology |
| ☐ | ☒ Animals and other organisms |
| ☒ | ☐ Human research participants |
| ☒ | ☐ Clinical data |
| ☒ | ☐ Dual use research of concern |

## Methods

| n/a | Involved in the study |
|---|---|
| ☒ | ☐ ChIP-seq |
| ☐ | ☒ Flow cytometry |
| ☒ | ☐ MRI-based neuroimaging |

# Antibodies

| | |
|---|---|
| Antibodies used | For immunofluorescence and immunohistochemistry the following antibodies were used: Rhodamine-labeled DBA (RL-1032, Vector Laboratories, 1:100), rhodamine-labeled UEA I (RL-1062, Vector Laboratories, 1:100), ZSGreen (mouse monoclonal, TA180002, Origene, 1:1000), CD31 (rabbit polyclonal, 77699, Cell Signaling Tehcnology, 1:100), CK7 (rabbit monoclonal [EPR17078],ab181598, Abcam, 1:200), CK20 (rabbit monoclonal,ab97511, Abcam, 1:200), E-cadherin (goat polyclonal, AF748, R&D systems, 1:100), N-cadherin (rabbit monoclonal, 13116, Cell Signalling Technology, 1:100), Icam1 (rabbit monoclonal, ab179707, Abcam, 1:100), Cdx2 (rabbit monoclonal, ab76541, Abcam, 1:100), aSMA (rabbit polyclonal, ab5694; Abcam; 1:500), Krt80 (rabbit polyclonal, 16835-1-AP, ProteinTech, 1:100), Hba-a1 (rabbit monoclonal, ab92492, Abcam, 1:100), Galectin3 (Lgals3) (rabbit monoclonal, ab209344, Abcam, 1:200), CD45 (rabbit polyclonal, ab10558, Abcam, 1:200), CD45 (ab25386, Abcam), Cleaved Caspase 3 (9664, Cell Signaling Technology, 1:200), Ki67 (IHC-00375, Bethyl, 1:1000). Following washing, tissue sections where then incubated for 1 hour at room temperature in secondary antibody. Secondary antibodies included Alexa 488, 594, 647 (A-11055, A-21207, A-31571, ThermoFisher, 1:500) of polymer based reagents (Leica Plymer Refine Detection System (DS9800, Leica biosystems) and Rabbit anti-rat polymer (A110-322a, Bethyl Laboratories, 1:250). Sections were then counterstained using DAPI (4083; Cell Signaling, 1:10,000) or Hematoxylin. |
| Validation | Abcam antibodies that are KO validated by the distributor: Lgals3 (rabbit monoclonal, ab209344, Abcam, 1:200). No knockout validation was carried out for any remaining antibodies..All other validation was done on primary tissues where the expected profile of the antibody was negative (e.g. Cdx2 expressed in GI tract but not lymphoid tissue) and also ensuring the correct compartmentalization of the antibodies (e.g. Cdh1 membrane bound) For over, for each of the antibodies when first using we would carryout no primary control histology to identify any background staining patterns and compared to slides that received the primary antibody. |

# Eukaryotic cell lines

Policy information about cell lines

| | |
|---|---|
| Cell line source(s) | Primary mouse gastric spheres generated from P1-KP gastric adenocarcinomas and P1-N normal gastric tissue., 293FT, L wnt3a |
| Authentication | None of the primary cell lines have been authenticated. The 293FT and L wnt3a were puchased from Thermo Fisher and ATCC, respectivley. |
| Mycoplasma contamination | The cell lines were tested prior to implantation into recipient immunocompromised animals and were negative. |
| Commonly misidentified lines (See ICLAC register) | None of the cell lines used are listed in the databased of commonly misidentified cell lines maintained by ICLAC. |

# Animals and other organisms

Policy information about studies involving animals; ARRIVE guidelines recommended for reporting animal research

| | |
|---|---|
| Laboratory animals | All genetically engineered mouse models were mixed background. Animals carrying the modified Nalcn allele were bred to RosaFLPe expressing mice to remove LacZ and Neo cassette. Animals with complete recombination were used for subsequent breeding combinations of various previously described strains: Prom1C-L; Nestin-cre; Rosa-CreERT2; villin-CreER; Pdx1-cre; RosaZSG; KrasG12D/+, Trp53flx were interbred and maintained on mixed Bl6/FVBN backgrounds. Both males and females were used in these experiments. Induction of recombination occurred either spontaneously embryonically (Nestin-cre, Pdx1-cre) or were induced postnatally ( day 3 or day 60) with tamoxifen and aged up to 2 years. For grafting experiments NSG/Foxn1nu/nu mice were used with age ranging from 4- 16 weeks. Female mice were only used in these experiments. Mice were housed in individually ventilated cages with wood chip bedding and nestlets with environmental enrichment (cardboard fun tunnels and chew blocks) in 12 h light/dark cycle at 21 ± 2 °C, humidity = 55% ± 10%. Diet was irradiated LabDiet 5R58 with ad libitum water. |
| Wild animals | study did not involve wild animals |
| Field-collected samples | study did not involve samples collected from the field |
| Ethics oversight | All animal studies within the United Kingdom were performed under the Animals (Scientific Procedures) Act 1986 in accordance with UK Home Office licenses (Project License 70-8823, P47AE7E47, PP7834816) and approved by the Cancer Research UK (CRUK) |

Cambridge Institute Animal Welfare and Ethical Review Board. All studies conducted at St. Jude Children's Research Hospital were approved by the Institutional Animal care and Use Committee of St. Jude Children's Research Hospital.

Note that full information on the approval of the study protocol must also be provided in the manuscript.

# Flow Cytometry

## Plots

Confirm that:

☒ The axis labels state the marker and fluorochrome used (e.g. CD4-FITC).

☒ The axis scales are clearly visible. Include numbers along axes only for bottom left plot of group (a 'group' is an analysis of identical markers).

☐ All plots are contour plots with outliers or pseudocolor plots.

☒ A numerical value for number of cells or percentage (with statistics) is provided.

## Methodology

| | |
|---|---|
| Sample preparation | Peripheral blood (500ml-1ml) was harvested from mice at necropsy, 10ul of 0.5M EDTA was added to the blood to prevent clotting and subject to red blood cell lysis (420301, BioLegend). ZSGreen cells were quantified using MACSQuant Analyzer 10 Flow cytometer (Miltenyi biotech). ZSGreen+ (ZSG+) sorted and quantified using a BD FACSAria II Cell Sorter (BD Biosciences) and the BD Influx Cell Sorter (BD Biosciences) with excitation at 525/50nm (FITC) vs 614/50nm (PI). Non-tamoxifen induced mouse peripheral blood served as a negative control to set parameters. |
| Instrument | MACSQuant Analyzer 10  Flow cytometer (Miltenyi biotech)<br>BD FACSAria II Cell Sorter (BD Biosciences)<br>BD Influx Cell Sorter (BD Biosciences) |
| Software | MACSQuant (Miltenyi biotech) and FACSAria Software (BD Biosciences) |
| Cell population abundance | ZSGreen populations range from 0 events up to 10% of events across all cell populations assessed. |
| Gating strategy | Living cells were selected by forward scatter and side scatter. ZSGreen positive populations were determined by using tamoxifen induced animal tissues as negative controls to set parameters. |

☒ Tick this box to confirm that a figure exemplifying the gating strategy is provided in the Supplementary Information.

